# Improving Sparse Vector Technique with Renyi Differential Privacy

**Yuqing Zhu**
Department of Computer Science
UC Santa Barbara
CA 93106
yuqingzhu@ucsb.edu

**Yu-Xiang Wang**
Department of Computer Science
UC Santa Barbara
CA 93106
yuxiangw@cs.ucsb.edu

## Abstract

The Sparse Vector Technique (SVT) is one of the most fundamental algorithmic tools in differential privacy (DP). It also plays a central role in the state-of-the-art algorithms for adaptive data analysis and model-agnostic private learning. In this paper, we revisit SVT from the lens of Renyi differential privacy, which results in new privacy bounds, new theoretical insight and new variants of SVT algorithms. A notable example is a Gaussian mechanism version of SVT, which provides better utility over the standard (Laplace-mechanism-based) version thanks to its more concentrated noise. Extensive empirical evaluation demonstrates the merits of Gaussian SVT over the Laplace SVT and other alternatives, which encouragingly suggests that using Gaussian SVT as a drop-in replacement could make SVT-based algorithms more practical in downstream tasks.

## 1 Introduction

The Sparse Vector Technique (SVT) [Dwork et al., 2009] is a fundamental tool in differential privacy (DP) that allows the algorithm to screen potentially an unbounded number of adaptively chosen queries while paying a cost of privacy only for a small number of queries that passes a predefined threshold.

SVT is the workhorse behind the *private multiplicative weights* mechanism [Hardt and Rothblum, 2010] and *median oracle* mechanism [Roth and Roughgarden, 2010], which famously shows that one can answer exponentially more linear queries differential privately for low-dimensional problems. It is also the key technique underlying the (conjectured optimal) improvements to the *ReusableHoldout* algorithms for preserving statistical validity in *adaptive data analysis* [Dwork et al., 2015b] and the *Ladder* algorithm for reliable machine learning leaderboards [Blum and Hardt, 2015]. We refer readers to the excellent course [Smith and Roth, 2017, Lecture 12] and the references therein.

More recently, SVT is combined with the *Distance to Stability* argument to build a machinery for *model agnostic private learning* in the knowledge transfer framework [Bassily et al., 2018]. The proposed algorithm releases many private labels from an ensemble of "teacher" classifiers trained on the private dataset [Bassily et al., 2018] while essentially only paying a privacy cost for those that are unstable. This in principle would allow the use any deep neural networks as a blackbox while leveraging the high-margin of the learned representation.

Despite the substantial benefit of SVT in theory, it is not known as a practical method. For example, in the case of model-agnostic private learning, SVT is often outperformed by simple Gaussian Mechanism [Papernot et al., 2018] that release all labels, since the latter uses a more concentrated noise (Gaussian over Laplace) and also has a tighter composition via Concentrated / Renyi differential privacy (CDP/RDP) [Dwork and Rothblum, 2016, Bun and Steinke, 2016, Mironov, 2017].

In this paper, we revisit SVT and address the following questions:

1. Is it essential to add Laplace noise? Does Gaussian noise work too? How about other noises e.g., [Geng and Viswanath, 2014]?

2. Is there a tighter RDP bound for SVT? Can we parameterize the RDP of SVT by the RDP function of the randomized mechanisms that are used to perturb the threshold and the answer to each query?

3. So far, the advanced composition of SVT is only available for the case when we compose $c$ SVTs with cut-off $= 1$, which requires refreshing the threshold noise each time. Could there be an $\sqrt{c}$ composition-theorem for the more general version when $c > 1$?

4. Finally, can we achieve better utility of SVT in practice? How small does $c$ needs to be relative to the total number of queries $k$ before SVT can outperform naive Gaussian mechanism?

5. Are there more practical alternatives to SVT that operates in those regimes where SVT fails.

We answer affirmatively to the first three questions (with some caveats and restrictions) by studying a generalized family of SVT (see Algorithm 2). Then we conduct numerical experiments to illustrate the pros and cons of various algorithms while highlighting the challenges in the last two questions. Moreover, we applied our results to the problem of adaptive data analysis and provided a "high probability" bound on the maximum accuracy of a sequence of $k$ adaptively chosen queries based on a Gaussian-mechanism variant of SVT, which matches (but unfortunately not improving) the strongest bound known to date on this problem.

**A remark on our novelty.** We believe our technical analysis that derives the RDP bound is new and elegant. Also our empirical evaluation is by far the more extensive for SVT-like algorithms. That said, we do borrow ideas from various prior work including [Lyu et al., 2017, Smith and Roth, 2017, Hardt and Rothblum, 2010] for the analysis including a cute trick from [Bun and Steinke, 2016], as well as getting practical insight and inspiration from [Papernot et al., 2018]'s data-dependent analysis of *noisy-screening*. A recent work [Liu and Talwar, 2019] generalized SVT to beyond low-sensitivity queries but still uses Laplace noise. We are different in that we develop SVT with other noise-adding mechanisms. Our technique should be directly applicable to the BetweenThreshold variant as in [Bun et al., 2017] an also release the "gap" as in [Ding et al., 2019]. The overarching goal of the paper is to make progress in bringing an amazing theoretical tool to practice. The improvements might be a constant factor in certain regimes but as differential privacy transitions into a practical technology, "constant matters!"

**Symbols and notations.** Throughout the paper, we will use standard notations for probability unless otherwise stated, e.g., $\Pr[\cdot]$ for probability, $p[\cdot]$ for density, $\mathbb{E}[\cdot]$ for expectation. Conditional probabilities, density and expectations are denoted with the standard $|$ in the middle, e.g., $\mathbb{E}[\cdot|\cdot]$, except for the cases when we state upfront that they abbreviated for lighter notations in that section. We do not distinguish fixed parameters and random variables as they are clear from context. The randomness are entirely the randomness induced by the randomized algorithm, except in the last section when we talk about adaptive data analysis. $\epsilon, \delta$ are reserved for privacy budget/loss parameters, and $\alpha$ the order of RDP. Other notations will be defined on the fly as they first appear.

## 2  Preliminary

In this section, we review the technical tools we use in this paper, introduce the sparse vector technique and highlight two applications of SVT. We start by formally defining differential privacy.

### 2.1  Differential privacy: pure, approximate and Renyi

**Definition 1** (Differential privacy[Dwork et al., 2006])**.** *A randomized algorithm $\mathcal{M}$ is $(\epsilon, \delta)$- differentially private (DP) if for any pair of neighboring dataset $D$ and $D'$, and any $S \subset Range(\mathcal{M})$,*

$$\Pr[\mathcal{M}(D) \in S] \leq e^{\epsilon} \cdot \Pr[\mathcal{M}(D') \in S] + \delta.$$

The definition provides rigorous guarantees by requiring the indistinguishability of whether or not a record is in the database based on the released information. $\epsilon, \delta \geq 0$ are privacy loss parameters. When $\delta = 0$, we have $\epsilon$-DP, or Pure DP.

**Algorithm 1** Standard SVT

**Input:** Data $D$, an adaptive sequence of queries $q_1, q_2, ... \in \mathcal{Q}$ with sensitivity $\triangle$, privacy parameter $\epsilon_1, \epsilon_2$, threshold $T$, cut-off $c$, option RESAMPLE.

1: Sample $\rho \sim \mathsf{Lap}(\triangle/\epsilon_1), \text{count} = 0$
2: for $i = 1, 2, 3, ...$
3:    Sample $\nu_i \sim \mathsf{Lap}(2\triangle/\epsilon_2)$
4:    **if** $q_i(D) + \nu_i \geq T_i + \rho$ **then**
5:      **Output** $a_i = \top, \text{count} = \text{count} + 1$
6:      if RESAMPLE, $\rho \sim \mathsf{Lap}(\triangle/\epsilon_1)$.
7:      if $\text{count} \geq c$, **abort**.
8:    **else**
9:      **Output:** $a_i = \perp$
10: **end if**

**Algorithm 2** Generalized SVT

**Input:** Data $D$, an adaptive sequence of queries $q_1, q_2, ... \in \mathcal{Q}$ with sensitivity $\triangle$, noise-adding mechanisms $\mathcal{M}_\rho, \mathcal{M}_\nu$, threshold $T$, cut-off $c$, max-length $k_{\max}$[a], option RESAMPLE.

1: Sample $\hat{T} \sim \mathcal{M}_\rho(D, T), \text{count} = 0$
2: for $i = 1, 2, 3, ..., k_{\max}$
3:    Sample $\hat{q}_i \sim \mathcal{M}_\nu(D, q_i)$
4:    **if** $\hat{q}_i \geq \hat{T}$ **then**
5:      **Output** $a_i = \top, \text{count} = \text{count} + 1$
6:      if RESAMPLE, $\hat{T} \sim \mathcal{M}_\rho(D, T)$
7:      if $\text{count} \geq c$, **abort**.
8:    **else**
9:      **Output:** $a_i = \perp$
10: **end if**

[a]You may choose $k_{\max} = \infty$ in some cases.

**Definition 2** (Renyi Differential Privacy [Mironov, 2017]). *We say a randomized algorithm $\mathcal{M}$ is $(\alpha, \epsilon(\alpha))$-RDP with order $\alpha \geq 1$ if for neighboring datasets $D, D'$*

$$\mathbb{D}_\alpha(\mathcal{M}(D)||\mathcal{M}(D')) := \frac{1}{\alpha - 1} \log \mathbb{E}_{o \sim \mathcal{M}(D')} \left[ \left( \frac{\Pr[\mathcal{M}(D) = o]}{\Pr[\mathcal{M}(D') = o]} \right)^\alpha \right] \leq \epsilon(\alpha).$$

Instead of considering each fixed RDP order in isolation, we take the functional view of RDP and use $\epsilon_{\mathcal{M}}(\cdot)$ to denote the RDP loss as a function of $\alpha$ (for $\alpha > 1$). This view unifies various popular notion of DP. At the limit of $\epsilon_{\mathcal{M}}(\infty)$, this recovers pure-DP. At the limit of $\alpha \to 1$, this recovers Kullback-Liebler DP (a.k.a. KL-stability). Moreover, we can covert RDP to $(\epsilon, \delta)$-DP for any $\delta > 0$ using:

**Lemma 3** (From RDP to DP). *If a randomized algorithm $\mathcal{M}$ satisfies $(\alpha, \epsilon(\alpha))$-RDP, then $\mathcal{M}$ also satisfies $(\epsilon(\alpha) + \frac{\log(1/\delta)}{\alpha - 1}, \delta)$-DP for any $\delta \in (0, 1)$.*

RDP is also connected to variants of the Concentrated Differential Privacy (CDP) [Dwork and Rothblum, 2016, Bun and Steinke, 2016], which can be thought of as a linear upper bound of the RDP function $\epsilon(\alpha)$. One convenient result that comes out of the CDP literature is the following RDP bound of a pure-DP mechanism.

**Lemma 4** (From DP to RDP [Bun and Steinke, 2016]). *Let $\mathcal{M}$ satisfy $\epsilon$-DP, then $\mathcal{M}$ also obeys $(\epsilon(\alpha), \alpha)$-RDP with $\epsilon(\alpha) = \frac{1}{\alpha - 1} \log \left( \frac{\sinh(\alpha\epsilon) - \sinh((\alpha - 1)\epsilon)}{\sinh(\epsilon)} \right) \leq \frac{\alpha\epsilon^2}{2}$.*

The $\sinh$ bound appears in the proof of Proposition 3.2 in [Bun and Steinke, 2016], which states the final bound, which means $\epsilon^2/2$-CDP. RDP satisfies many information-theoretic properties of DP including composition, closure to post-processing and so on (see Appendix A for more details.)

The most common mechanisms for differential privacy are those that add noise to queries answers.

**Definition 5** (Noise-adding mechanisms). *We say that $\mathcal{M} : Data \times \mathcal{Q} \to P_\mathbb{R}$ is a noise-adding mechanism if it answers a query $q$ by outputting $o \sim \mathcal{M}(D, q) = q(D) + Z$ where $Z$ is a random variable.*

Typical examples of these noise-adding mechanisms for differential privacy includes Laplace-mechanism, Gaussian mechanism in which $Z$ is drawn from a Laplace distribution and a Gaussian distribution respectively. Notably, the "optimal" geometric mechanism falls under this category which adds a "stair-case"-shape noise [Geng and Viswanath, 2014].

**Definition 6** (Low-sensitivity queries). *We define $\mathcal{Q}(\triangle)$ to be the set of all queries $q : Data \to \mathbb{R}$ such that $|q(D) - q(D')| \leq \triangle$ for any pair of neighboring datasets $D, D'$.*

$\triangle$ is called global sensitivity and is used to calibrate the noise according to a given privacy budget.

## 2.2 Sparse vector techniques

In SVT, the input is a stream of possibly infinitely long, adaptively chosen queries $q_1, q_2, ..., q_i, ... \in \mathcal{Q}(\triangle)$. The queries are provided with a sequence of thresholds $T_1, T_2, ..., T_k, ...$. The goal of SVT is

to release a binary vector $\{\bot, \top\}^k$ at every time $k$, $\top$ indicates that the corresponding query answer $q_i(D)$ is above the threshold $T_i$ and $\bot$ indicates below. To release this vector differential privately, we first perturb the threshold $T$ with a Laplace noise $\rho$. Then each individual query $q_i(D)$ is perturbed by another Laplace noise $\nu_i$ before comparing against the perturbed threshold $T + \rho$ to determine the binary decion, until the stopping condition — the c-th $\top$ arrives. Algorithm 1 summarizes pseudo-code from [Hardt and Rothblum, 2010] and [Lyu et al., 2017].

A remarkable property of SVT is that it allows the release of a vector that is exponentially long while incurring only a privacy loss proportional to $c$ (or its square root) — the maximum number of answers that are allowed to be $\top$. This is formalized in the following lemma.

**Lemma 7** (Privacy calibration in Standard SVT). *Algorithm 1 satisfies* $(\epsilon_1 + c\epsilon_2)$-*DP when* RESAMPLE *option is set to false. When* RESAMPLE = True, *then Algorithm 1 with* $\frac{\triangle}{\epsilon_1} = \frac{\triangle}{\epsilon_2} = \frac{\sqrt{32c\log(1/\delta)}}{\epsilon}$, *then Algorithm 1 satisfies* $(\epsilon, \delta)$-*DP and* $(c\epsilon_1 + c\epsilon_2)$-*DP.*

The version of the pure-DP calibration without resampling comes from [Lyu et al., 2017, Algorithm 1]. The $(\epsilon, \delta)$-DP calibration is extracted from [Dwork and Roth, 2013, Theorem 3.25], which is essentially applying strong composition to to $c$ instances of SVT, each obeying $(\epsilon_1 + \epsilon_2)$-DP.

Despite the asymptotic savings from $\sqrt{k}$ to $\sqrt{c}$, SVT is still not known as a practical mechanism. The reasons, in our opinion, are twofolds.

The Laplace distribution used in the SVT is a heavy-tailed (sub-exponential) distribution, which requires the threshold to be set to $O(\log(1/\beta))$ as to control the false positive rate at $\beta$. This could be much larger than the $O(\sqrt{\log(1/\beta)})$ of sub-gaussian tailed distributions, hence make SVT less favorable for utility-privacy trade-off in practice. Moreover, many practical differential private algorithms benefit from tighter privacy accounting, e.g., composition using Renyi DP with numerical computation. It will be ideal if we can come up with a version of the SVT that adds more concentrated noise as well as a general Renyi DP analysis of that algorithm. This motivated us to consider the family of Generalized SVT mechanism in Algorithm 2.

## 3 Main results

In this section, we derive RDP bounds for SVT variants with different distributions of noisy parameters $(\nu_i, \rho)$. The goal is to find those distributions that not only have thin tail bounds but preserve the essential property of the standard SVT — they can answer exponentially many $\bot$ queries while paying a privacy loss only proportional to $c$ or $\sqrt{c}$ when the algorithm halts. The family of mechanisms we consider is summarized in Algorithm 2. The differences from Algorithm 1 are highlighted in blue.

### 3.1 RDP analysis with $c = 1$

We first consider generalized SVT with $c = 1$, since the case of $c > 1$ is often treated as composition of multiple SVT with $c = 1$.

**Theorem 8.** *Let $K$ be a random variable indicating the stopping time — number of $\bot$s plus 1. Let $\mathcal{M}_\rho$, $\mathcal{M}_\nu$ be noise-adding mechanisms (Definition 5). Assume $\mathcal{M}_\rho$ satisfies $\epsilon_\rho(\alpha)$-RDP for queries with sensitivity $\triangle$ and $\mathcal{M}_\nu$ satisfies $\epsilon_\nu(\alpha)$-RDP for queries with sensitivity $2\triangle$. Then Algorithm 2 with $c = 1$ (denoted by $\mathcal{M}$) obeys*

$$\mathbb{D}_\alpha(\mathcal{M}(D)\|\mathcal{M}(D')) \leq \epsilon_\rho(\alpha) + \epsilon_\nu(\alpha) + \frac{\log \sup_z \mathbb{E}[K|\rho = z]}{\alpha - 1}, \quad (1)$$

$$\mathbb{D}_\alpha(\mathcal{M}(D)\|\mathcal{M}(D')) \leq \frac{\alpha - (\gamma - 1)/\gamma}{\alpha - 1}\epsilon_\rho\left(\frac{\gamma}{\gamma - 1}\alpha\right) + \epsilon_\nu(\alpha) + \frac{\log\left(\mathbb{E}_{z \sim p_\rho}[\mathbb{E}[K|\rho = z]^\gamma]\right)}{\gamma(\alpha - 1)}, \quad (2)$$

*for all $\gamma > 1$ and $1 < \alpha < \infty$. Moreover, when $\epsilon_\rho(\infty) \leq \infty$, we get*

$$\mathbb{D}_\alpha(\mathcal{M}(D)\|\mathcal{M}(D')) \leq \epsilon_\rho(\alpha) + \epsilon_\nu(\infty). \quad (3)$$

The theorem can be thought of as a general transfer theorem that allows us to bound the RDP of the generalized SVT with the RDP of its subroutines $\mathcal{M}_\rho$ and $\mathcal{M}_\nu$. Before proving the theorem in Section B, let us parse the result in a number of special cases.

**Remark** (Pure-DP). RDP (3) recovers the pure-DP bound of the standard SVT when $\alpha \to \infty$. It also allows other noise-adding procedure that satisfies pure-DP to be applied. We could also consider the hybrid-noise SVT where $\rho$ is a Gaussian noise, but $\nu$ are Laplace-noises.

**Remark** (Bounded-length SVT). When we set $k_{max} < +\infty$, the (1) implies an RDP bound of the form $\epsilon_\rho(\alpha) + \epsilon_\nu(\alpha) + \log(1 + k_{max})/(\alpha - 1)$, which further implies an $(\epsilon, \delta)$-DP bound by Lemma 3. In particular, if $\delta \leq 1/(1 + k_{max})$, then we get $(\epsilon, \delta)$-DP with

$$\epsilon = \min_{\alpha > 1} \epsilon_\rho(\alpha) + \epsilon_\nu(\alpha) + 2\log(1/\delta)/(\alpha - 1).$$

In the case of Gaussian mechanism, this loses at most a factor of $\sqrt{2}$ comparing to the case when $\log(1 + k_{max})/(\alpha - 1)$ is not there all together.

**When $k_{max}$ is chosen to be $+\infty$:** (1) and (2) do not imply RDP in this case, because there are cases where SVT can potentially have unbounded length (in fact, even the expected length can be unbounded[1] ). It is well-expected that if we use Gaussian-mechanism as a subroutine for SVT, the dependence of the sequence length is unavoidable. Similar observations have been made about a Gaussian-noise version of the ReportNoisyMax mechanism [see, e.g., Dwork and Roth, 2013, Section 3.5.3]. That said, the form of the bound (2), which depends only on the moments of the conditional expectation seems to suggest that we can potentially obtain meaningful RDP bounds for generalized SVT even if $k_{max} = +\infty$ in some cases.

Let us consider a mild restriction to the family of queries that can be chosen, which allows us to keep the sequence length unbounded even when the noise-adding subroutines do not satisfy pure-DP.

**Definition 9** (Nonnegative, Low-sensitivity Queries Model). *The adversary can adaptively choose* $q_1, q_2, ... \in \mathcal{Q}_+(\triangle)$ *where*

$$\mathcal{Q}_+(\triangle) = \{q : Data \to \mathbb{R} \mid q(D) \geq 0 \forall D, \ |q(D) - q(D')| \leq \triangle \forall \text{ neighboring datasets } D, D'\}.$$

The class covers both use cases of SVT that we described earlier. When we apply SVT to "Guess-and-Check"[2], $q_i(D) = \|f_i(D) - g_i\|$ is nonnegative. Similarly, in the case of "Model-agnostic private learning" $q_i(D) = \text{dist}_{\text{MajorityVote}_{f_i}}(D)$, which measures the number of data points that need to be added or removed to make the argmax of the voting score unstable.

**Proposition 10** (Gaussian SVT with non-negative queries). Let Algorithm 2 be instantiated with $\mathcal{Q}_+(\triangle)$, $\mathcal{M}_\rho$ and $\mathcal{M}_\nu$ be Gaussian mechanism with parameter $\sigma_1$ and $\sigma_2$. Then for all $T < +\infty$ and $\gamma > 1$ such that $\sigma_2 > \sqrt{\gamma + 1}\sigma_1$, Algorithm 2 with $c = 1$ halts with $K$ rounds satisfying

$$\mathbb{E}[\mathbb{E}[K|\rho = z]^\gamma] \leq 1 + (c_\gamma \sqrt{2\pi} \max\{\frac{T(1+\gamma)}{\sigma_1}, 1\})^\gamma (1+\gamma)^{1/2} e^{\frac{\gamma T^2}{2\sigma_1^2}}.$$

For Gaussian SVT satisfying $\sigma_2 \geq \sqrt{3}\sigma_1$, it obeys an RDP of $\frac{\alpha \triangle^2}{\sigma_1^2} + \frac{2\alpha \triangle^2}{\sigma_2^2} + \frac{\log(1 + 2\sqrt{3}\pi(1 + \frac{9T^2}{\sigma_1^2})e^{\frac{T^2}{\sigma_1^2}})}{2(\alpha - 1)}$.

The proof of Proposition 10, deferred to the appendix, hinges upon the key observation that $K$ follows a Negative Binomial distribution when conditioning on the threshold, and some technical calculations involving Mill's ratio and moments of Gaussian distribution.

**Remark** (Controlling Type I error). One can for example choose $T = \sqrt{2(\sigma_1^2 + \sigma_2^2)\log 1/\varrho} = \sqrt{8\sigma_1^2 \log(1/\varrho)}$ such that the Type I error (false positive rate) is bounded by $\varrho$. This is often the case when using sparse vector technique for statistical applications. Then we can simplify the above bound by using $\log(1 + x) \leq x$, and an assumption that $\rho$ is sufficiently small, to obtain an RDP bound of

$$\frac{5\alpha \triangle^2}{3\sigma_1^2} + \frac{8\log(1/\rho) + \log(4\sqrt{3}\pi(1 + 72\log(1/\rho)))}{2(\alpha - 1)} \leq \frac{5\alpha \triangle^2}{3\sigma_1^2} + \frac{5\log(1/\rho)}{(\alpha - 1)}.$$

The above results allow us to obtain nearly the same $(\epsilon, \delta)$-DP bound for Gaussian-SVT as if we are working with the RDP bound of a Gaussian mechanism, provided that the $\delta$ chosen such that

$\log(1/\delta)$ is larger than either $\log k_{\max}$ in the length-bounded case or $O(\log(1/\rho))$ in the nonnegative query setting.

To ease our subsequent presentation, from here onwards we will use $k_\gamma$ to denote a data-independent upper bound of $(\mathbb{E}[K|\rho]^\gamma])^{1/\gamma}$. Conveniently, $k_\infty = k_{\max}$ and $k_1$, which unifies (1) and (2). Moreover, we will use $\gamma^*$ such that $1/\gamma^* + 1/\gamma = 1$.

## 3.2 Generalized SVT with $c > 1$

We now address the case when $c > 1$. A natural, and general way to deal with SVT for $c > 1$ is to simply apply composition theorems of differential privacy to $c$ instances of SVT with cut-off parameter $c$ set to 1. We could also directly analyze the variant of SVT with $c > 1$, where the threshold noise is not refreshed. Pros and cons of these two approaches are described in Appendix D.

**Theorem 11** (RDP for length-capped SVT with $c > 1$). *The generalized SVT with cut-off parameter $c > 1$ and a maximum length is $k_{\max}$ obeys that*

$$\mathbb{D}_\alpha(\mathcal{M}(D)\|\mathcal{M}(D')) \le \epsilon_\rho(\alpha) + c\epsilon_\nu(\alpha) + \frac{1 + \log \sum_{k=0}^c \binom{k_{\max}}{k}}{\alpha - 1}.$$

The proof, presented in the Appendix, uses the same techniques as in the proof of Theorem 8, but we no longer get an interpretable bounds that rely on moments of $\mathbb{E}[K|\rho]$. The term in the logarithmic factor, resembles $k_{\max}$ in the sense that it counts the cardinality of the output space — binary vectors of length $k_{\max}$ with at most $c$ $\bot$s.

**Remark.** When both noise are Gaussian, the theorem and Lemma 3 implies an $(\epsilon, \delta)$-DP with

$$\epsilon(\delta) \le \frac{\Delta^2}{2\sigma_1^2} + \frac{2c\Delta^2}{\sigma_2^2} + \sqrt{2\left(\frac{\Delta^2}{2\sigma_1^2} + \frac{2c\Delta^2}{\sigma_2^2}\right)\left(\log(\delta^{-1}) + \log c\binom{k_{\max}}{c}\right)}$$

which recovers the $O(\sqrt{c})$ scaling when $\delta \le c^{-1}\binom{k_{\max}}{c}^{-1}$ and saves a factor of $c$ in $\sigma_1$.

While the restriction on $\delta$ being smaller than $k_{\max}^{-c}$ is quite limiting, we are not aware of an analysis that achieves the strong composition-like scaling in $c$ for the version of the SVT that does not refresh the noise under any parameter configurations.

**Back to $(\epsilon, \delta)$-composition.** Interestingly, if we use of the strong composition for $(\epsilon, \delta)$-DP directly, we can obtain a bound with $\sqrt{c}$ scaling for a much broader set of parameters. Let us consider the following stage-wise algorithm for Generalized SVT, which resamples the threshold noise $\rho$ after every $c'$ rounds with a pre-specified bound $k'_{\max}$ chosen in each round. This algorithm can be viewed as a meta-algorithm that calls Algorithm 2 as a subroutine (see Algorithm 3). The idea is that we can choose $c'$ and $k'_{\max}$ carefully according to $c$ and $\delta$ such that for each call of SVT, the region of interests falls under the region where $\log(c'\binom{k'_{\max}}{c'})$ is comparable to $\log(1/\delta)$.

**Theorem 12** ( Stage-wise Length-Capped Gaussian SVT for $\mathcal{Q}(\triangle)$). *Let $0 < \delta' < 1$ be a parameter. Let $\mathcal{M}$ be the instance of the Algorithm 3 invoked with cut-off $c'$, max-length $k'$, option $\mathsf{RESAMPLE} = \mathsf{False}$, $\mathcal{M}_\rho, \mathcal{M}_\nu$ chosen as Gaussian mechanisms with noise parameter $\sigma_1, \sigma_2$ satisfying $\sigma_2 = 2\sigma_1$ and $\sigma_1 \ge 8\triangle\sqrt{c\log(1/\delta')}$. If we choose $c' \le c$ such that $c'\binom{k'_{\max}}{c'} \le (\delta')^{-1}$, then $\mathcal{M}$ satisfies $(\epsilon, \tilde{\delta} + \frac{c}{c'}\delta')$-DP with $\epsilon = O\left(\sqrt{\frac{c\triangle^2}{\sigma_1^2}\log(1/\delta')\log(1/\tilde{\delta})}\right)$.*

**Theorem 13** (Adaptive Stage-wise Gaussian SVT for $\mathcal{Q}_+(\triangle)$). *Let $\mathcal{M}$ be an instance of Algorithm 3 invoked with the same parameters as in Theorem 12, except that $\mathsf{RESAMPLE} = \mathsf{True}$ and $k_{\max} = +\infty$. Then for all $c'$, $\gamma$ such that $k_\gamma^{c'} \le (\delta')^{-1}$, then $\mathcal{M}$ is $(\epsilon, \tilde{\delta} + \frac{c}{c'}\delta')$-DP with $\epsilon = O\left(\sqrt{\frac{c\triangle^2}{\sigma_1^2}\log(1/\delta')\log(1/\tilde{\delta})}\right)$ for all adaptively chosen sequences of queries in $\mathcal{Q}_+$.*

**Remark** (Adaptive to $c'$ and numerical computation). Observe that choosing $\mathsf{RESAMPLE} = \mathsf{True}$ makes Algorithm 3 identical to Algorithm 2 for all choices of $c' \ge 1$. We can thus minimize the bound numerically over the parameters $c', \delta'$ to minimize the final bound, to simulate the conceptual process of which part of the composition is RDP-based and which part $(\epsilon, \delta)$-DP based. The best choice would be to make $c'$ as large as possible so as to get the partial benefit of the savings from RDP composition over the $c'$ steps within each stage, while not ruining the $O(\sqrt{c})$ strong composition when $c$ is large.

---

**Algorithm 3** Stage-wise generalized SVT

---

**Input:** Data $D$, an adaptive sequence of queries $q_1, q_2, ... \in \mathcal{Q}$ with sensitivity $\triangle$, noise-adding mechanisms $\mathcal{M}_\rho, \mathcal{M}_\nu$, threshold $T$, total cut-off $c$, per-stage cut-off $c'$, per-stage max-length $k'_{\max}$, option RESAMPLE.

1: Initialize output vector to be an empty list.
2: **for** for $\ell = 1, 2, 3, ..., \lceil c/c' \rceil$ **do**
3:     Set $\tilde{c} = c - c'(\lceil c/c' \rceil - 1)$ if $\ell = \lceil c/c' \rceil$, and $\tilde{c} = c'$ otherwise.
4:     Invoke Algorithm 2 with $D, T, \mathcal{M}_\rho, \mathcal{M}_\nu, \tilde{c}, k'_{\max}$, RESAMPLE and current front of the adaptive stream of queries.
5:     Append the new output vector from Algorithm 2 to the output.
6: **end for**

---

**Remark.** Comparing to the likely-unachievable conjecture where the generalized SVT has an RDP of $\epsilon_\rho(\alpha) + c\epsilon_\nu(\alpha)$, which would give an $\epsilon = O(\sqrt{\frac{c\triangle^2}{\sigma_1^2} \log(1/\delta)})$, this bound is worse only by a factor of $\sqrt{\log(1/\delta)}$, and has some mild restrictions on $\delta$. We remark that in Theorem 12 and 13 we focused on the asymptotic scaling, while in practice, we can use the optimal advanced composition due to [Kairouz et al., 2015] and search for the best parameters to give the tightest bounds.

### 3.3 Application to Adaptive Data Analysis

The stage-wise length-bounded Gaussian SVT's $(\epsilon, \delta)$-DP guarantee allows us to directly apply it to the problem of adaptive data analysis that aims at preventing data dredging while still allowing an analyst to get accurate answers about a sequence of $k$ adaptively chosen statistical queries through an interactive protocol [Dwork et al., 2015b, Smith, 2017].

**Theorem 14.** *With probability $\geq 1 - \delta$ over the random coins of the i.i.d. data, our algorithm and other randomness coming from the interaction protocols against an arbitrary adaptive adversary, the Gaussian-SVT-based Private-Guess-and-Check answers $k$ queries including at most $c$ inaccurately guesses with generalization error at most $O(\frac{c^{1/4} \log(k/\delta)^{3/4}}{n^{1/2}})$.*

The proof combines either Theorem 12 or 13 with the high probability generalization bound of $(\epsilon, \delta)$-DP algorithms [Jung et al., 2020] as well as the Gaussian tail bound.

In comparison, the simple Gaussian mechanism guarantees an accuracy of $O(k^{1/4} \log(k/\delta)^{1/2} n^{-1/2})$ and the original *ReusableHoldout* gives $O(c^{1/2}\sqrt{\log(k/\delta)}n^{-1/2})$. We show that Gaussian SVT improves over these and matches the best known rate for the problem achieved by Laplace-mechanism-base SVT — an $O(\log(k/\delta)^{1/4})$ away from the lower bound. Interestingly the reason of the suboptimality is different. Laplace SVT is off due to the subexponential tail bound of Laplace R.V., while Gaussian SVT is off due to the additional $O(\log(k/\delta)^{1/2})$ factor from the strong composition. It remains an open problem how to close this gap.

## 4 Experiment and discussion

In this section, we conduct extensive numerical experiments to illustrate the behaviors of SVT variants. We will have three sets of experiments.

**Exp. 1** (Calibrating noise to privacy) Given a predetermined privacy budget $(\epsilon, \delta)$ and the cut-off $c$, we compare the length each SVT-like algorithm can screen before stopping.

**Exp. 2** (Privacy loss computation) We evaluate SVT variants with the same variance of noise by comparing the composed privacy loss for finishing a fixed length sequence of queries.

**Exp. 3** (Real life data) We investigate various private screening methods with a realistic sequence of queries from running a kNN-based private-query release on the CIFAR-10 dataset.

In Exp 1 and Exp 2, the sequences of queries are $q_t(D) = 0$ for all $t$ so all discoveries that end up being detected as $\top$ are false positives. Thus the length of the sequence is a measure of utility in Exp 1. Exp 2 and Exp 3 compares the expected privacy loss $\epsilon$ at a fixed $\delta$ as it composes. For all experiments, we denote $(\sigma_1, \sigma_2)$ or $(\lambda_1, \lambda_2)$ as the noise to perturb threshold and query in Gaussian and Laplace, respectively. The ratio between the query noise and the threshold noise is fixed —

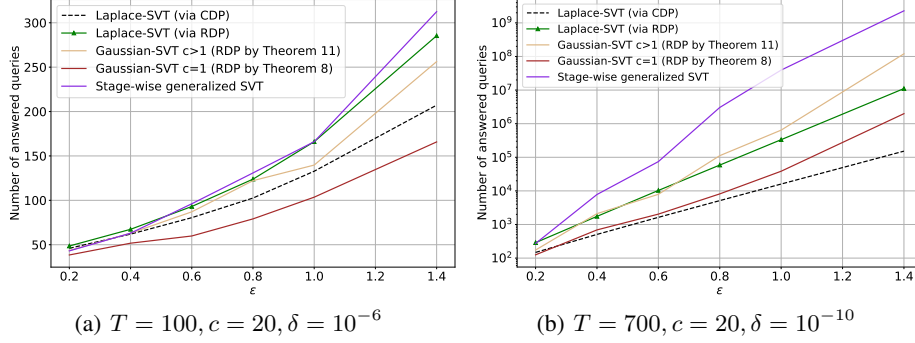

(a) $T = 100, c = 20, \delta = 10^{-6}$        (b) $T = 700, c = 20, \delta = 10^{-10}$

Figure 1: Number of queries each algorithm can process with a fixed privacy budget $(\epsilon, \delta)$, fixed cut-off (# of false positives) $c$ and fixed threshold $T$.

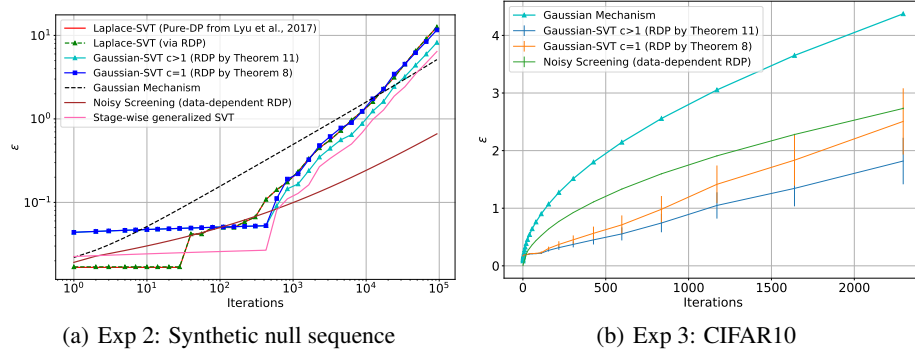

(a) Exp 2: Synthetic null sequence        (b) Exp 3: CIFAR10

Figure 2: Total composed privacy loss as the algorithm progresses for $\delta = 10^{-6}$. The margin $T = 1000$ and $\sigma_1 = 210$. The standard deviation of Gaussian and Laplace are aligned to be comparable.

$\frac{\sigma_2}{\sigma_1} = \frac{\lambda_2}{\lambda_1} = 2$. When applicable, we include simple Gaussian mechanism as a baseline. Moreover, we added noisy-screening, which basically output $\perp$ if $q_t(D) + \mathcal{N}(0, \sigma^2) \geq T$ and $\top$ otherwise. The data-dependent RDP-bound for noisy-screening [Papernot et al., 2018, Theorem 6] behaves like SVT as it pays exponentially smaller privacy loss when the query $q_t(D)$ gets far from the threshold $T$. We emphasize the privacy loss is sensitive information, thus not directly comparable to other DP methods. Finally for Gaussian-SVT, $k_{\max}$ needs to be chosen carefully. Due to space limit we defer more details about the experimental setup and more exposition to Appendix G.

**Observations on the experiments.** In Experiment 1, when the tail of the noise plays a significant role, e.g. the threshold T is large (Figure 1(b)), Gaussian-SVT is more advantageous due to a more concentrated noise. To further improve Gaussian-SVT, the stage-wise Gaussian-SVT that uses hybrid composition (Theorem 12) outperforms Laplace-SVT significantly. On a side note, the $\sinh$-style RDP bound for Laplace-SVT ($c = 1$) from Lemma 4 turns out to be quite a bit better than the CDP-version and the standard calibration ( Lemma 7). In Experiment 2, we see that as the privacy loss composes Laplace-SVT and Gaussian-SVT with the same noise variance behave qualitatively similar. Gaussian-SVT is better by a constant factor with larger number of iterations. Meanwhile, naive Gaussian mechanism and noisy-screening is often the better choice when the number of iterations is small. In Experiment 3, we see that the expected privacy losses of Gaussian SVT outperforms that of the noisy-screening despite that the latter is data-dependent. The error bars are computed based on 10 independent run and has a correct 95% coverage. We excluded Laplace-SVT in Exp 3 due to the lack of a way for fair comparison.

## 5 Conclusion

To conclude, we developed a generalization of sparse vector technique for DP that allows us to use any noise-adding mechanisms. We derived the Renyi-DP bounds of these generalized-SVT and showed that we can get $\sqrt{c}$-composition in all practical regimes of interests. We use theory and experiments to demonstrate the merits of Gaussian-SVT. In downstream tasks, we have shown that Gaussian-SVT matches the best existing bound for adaptive data analysis and demonstrated in experiments that it could improve the privacy accounting in model-agnostic private learning. We hope the work will spark new ideas and practical applications involving SVT.

## Broader impacts

As mentioned in the paper, differential privacy is undergoing an exciting transition from theory to practice and there is an increasing number of deployment of differential private algorithms and systems in both the private and public sectors.

The focus of the work is to bring a mature and classical technique in differential privacy — Sparse Vector Technique — to practice by improving the privacy-utility trade-off, and also to conduct numerical studies so as to provide recommendations on which method to use in each regime. This task is strongly tied to the goal of AI/ML for social good and responsible computing.

Notice that in practice, the computation of privacy loss or the calibration of noise to privacy budgets is extremely important as these seemingly theoretical calculations will affect the number of data points to collect, and affect the statistical power in sensitive applications such as clinical studies.

Moreover, our application to adaptive data analysis is strongly tied to the reproducibility crisis in science and the problems of overfitting common benchmarks that we are currently experiencing in the machine learning community.

### Acknowledgment

The research was partially supported by the start-up grant of YW at UCSB Computer Science and generous gifts from Amazon Web Services and NEC Labs.

## Footnotes

[1]see an example in the appendix.

[2]A subroutine of "private multiplicative weights" and "reusable holdout".

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
