[Supplementary Material]

# A  Other properties of differential privacy and RDP

RDP inherits and generalizes the information-theoretic properties of DP.

**Lemma 15** (Selected Properties of RDP [Mironov, 2017]). *If $\mathcal{M}$ obey $\epsilon_{\mathcal{M}}(\cdot)$-RDP, then*

1. *[Indistinguishability] For any measurable set $S \subset Range(\mathcal{M})$, and any neighboring $D, D'$*

$$e^{-\epsilon(\alpha)}\Pr[\mathcal{M}(D') \in S]^{\frac{\alpha}{\alpha-1}} \leq \Pr[\mathcal{M}(D) \in S] \leq e^{\epsilon(\alpha)}\Pr[\mathcal{M}(D') \in S]^{\frac{\alpha-1}{\alpha}}.$$

2. *[Post-processing] For all function $f$, $\epsilon_{f \circ \mathcal{M}}(\cdot) \leq \epsilon_{\mathcal{M}}(\cdot)$.*

3. *[Composition] $\epsilon_{(\mathcal{M}_1, \mathcal{M}_2)}(\cdot) = \epsilon_{\mathcal{M}_1}(\cdot) + \epsilon_{\mathcal{M}_2}(\cdot)$.*

This composition rule, together with Lemma 3, often allows for tighter calculations of $(\epsilon, \delta)$-DP for the composed mechanism than directly invoking the strong composition theorem below.

**Lemma 16** (Strong composition [Kairouz et al., 2015]). *For all $\epsilon, \delta, \delta' \geq 0$, the (adaptive) composition of $k$ $(\epsilon, \delta)$-DP mechanisms obey $(\epsilon', \delta' + k\delta)$-DP for $\epsilon' = \sqrt{2k \log(1/\delta')}\epsilon + k\epsilon\frac{e^{\epsilon}-1}{e^{\epsilon}+1}$.*

# B  Proof of Theorem 8 — RDP of the Generalized SVT for $c = 1$

For unbounded sequences, the output space of the algorithm is $\{\perp^k \top | k = 0, 1, ..., \infty\}$. In the case when $k_{\max} < +\infty$, the output space is $\{\perp^k \top | k = 0, 1, ..., k_{\max} - 1\} \cup \{\perp^{k_{\max}}\}$. For notation convenience, we replace $\perp^{k_{\max}}$ with $\perp^{k_{\max}} \top$, which can be thought of as fixing a dummy query at time $k_{\max} + 1$ which always outputs $+\infty$ regardless of inputs. In both cases, we can completely describe the output distribution the SVT with a positive random integer $K$. As a result, we will write $K \sim \mathcal{M}(D)$ and $K \sim \mathcal{M}(D')$ without loss of generality.

Also w.l.o.g., we assume thresholds $T_i$ are all zero. There are two types of random variables in the algorithm: the threshold noise $\rho$ and the query noise $\nu_i$ to each of the $i$ queries, $\{\nu_i\}_{i=1}^{k+1}$. We will use $p_{\rho}(z)$ to denote the probability density of $\rho$, evaluated at $z$, and we will use $p(\nu_i)$ as the pdf of $\nu_i$.

The probability of outputting $o$ (or $K = k + 1$), can be written explicitly as follows:

$$\Pr[\mathcal{M}(D') = o] = \int_{-\infty}^{+\infty} p_{\rho}(z)\left(\prod_{i \leq k} \int_{-\infty}^{z - q_i(D')} p(\nu_i)d\nu_i\right) \cdot \int_{z+q_{k+1}(D')}^{\infty} p(\nu_{k+1})d\nu_{k+1}dz.$$

Our goal of is to bound $\mathbb{E}_{o \sim \mathcal{M}(D')}\left[\left(\frac{\Pr[\mathcal{M}(D)=o]}{\Pr[\mathcal{M}(D')=o]}\right)^{\alpha}\right]$ using the RDP functions of $\mathcal{M}_{\rho}$ and $\mathcal{M}_{\nu}$.

The key of the analysis relies on a sequence of fictitious queries $\tilde{q}_1, \tilde{q}_2, ...$ which mirrors the actual sequence of queries $q_1, q_2, ...$ that are adaptively selected. These fictitious queries satisfy for all $i = 1, 2, 3, ...$

$$\tilde{q}_i(x) = \begin{cases} q_i(D) + \triangle, & \text{when } x = D \\ q_i(D') & \text{otherwise} \end{cases} \tag{4}$$

The following lemma establishes that we can decompose the problem into one that involves the Renyi-divergence between a distribution induced by these fictitious queries and another distribution induced of the actual queries.

**Lemma 17.** *Consider Algorithm 2 with $c = 1$, i.e., the output sequence $o \in \{\perp^k \top | k = 0, 1, ..., \infty\}$, then we have*

$$\mathbb{E}_{o \sim \mathcal{M}(D')}\left[\left(\frac{\Pr[\mathcal{M}(D) = o]}{\Pr[\mathcal{M}(D') = o]}\right)^{\alpha}\right] \leq \mathbb{E}_{z \sim p_{\rho}}\left[\left(\frac{p_{\rho}(z - \triangle)}{p_{\rho}(z)}\right)^{\alpha} \underbrace{\mathbb{E}_{K \sim \mathcal{M}(D')}\left[\frac{\left(\Pr[\mathcal{M}(D) = K|z, \tilde{\mathcal{Q}}]\right)^{\alpha}}{(\Pr[\mathcal{M}(D') = K|z])^{\alpha}}\middle| z\right]}_{\text{denoted by } (*)}\right]$$

$$\tag{5}$$

*where $K$ is a random variable, denotes the number of $\perp$ plus 1 when the algorithm stops, and the explicit conditioning on $\tilde{\mathcal{Q}}$ indicates that the probability is evaluated by hypothetically running the algorithm on the fictitious queries $\tilde{q}_1, \tilde{q}_2, ... \in \tilde{\mathcal{Q}}$.*

*Proof of Lemma 17.* From the definition of Renyi DP, we have

$$\mathbb{E}_{o\sim D'}\left[\frac{\Pr[\mathcal{M}(D)=o]^\alpha}{\Pr[\mathcal{M}(D')=o]^\alpha}\right]=\sum_{k=0}^\infty\frac{\Pr[\mathcal{M}(D)=\perp^k\top]^\alpha}{\Pr[\mathcal{M}(D')=\perp^k\top]^{\alpha-1}}\tag{6}$$

Without loss of generality, we will replace $o$ with $k$ which measures the number of $\perp$s in $o$. By law of total expectation, we can condition on $\rho=z$

$$\Pr[\mathcal{M}(D)=k]=\mathbb{E}_{z\sim p_\rho}[\Pr[\mathcal{M}(D)=k|z]]$$

$$=\mathbb{E}_{z\sim p_\rho}[\prod_{i\leq k}\Pr[q_i(D)+\nu_i<z|z]\Pr[q_{k+1}(D)+\nu_i\geq z|z]]$$

$$=\int_{-\infty}^{+\infty}p_\rho(z)\left(\prod_{i\leq k}\int_{-\infty}^{z-q_i(D)}p(\nu_i)d\nu_i\right)\cdot\int_{z-q_{k+1}(D)}^\infty p(\nu_{k+1})d\nu_{k+1}dz$$

$$\overset{u:=z+\triangle}{=}\int_{-\infty}^{+\infty}p_\rho(u-\triangle)\left(\prod_{i\leq k}\int_{-\infty}^{u-\triangle-q_i(D)}p(\nu_i)d\nu_i\right)\cdot\int_{u-\triangle-q_{k+1}(D)}^\infty p(\nu_{k+1})d\nu_{k+1}du$$

$$=\int_{-\infty}^{+\infty}p_\rho(u)\left(\frac{p_\rho(u-\triangle)}{p_\rho(u)}\right)\left(\prod_{i\leq k}\int_{-\infty}^{u-\triangle-q_i(D)}p(\nu_i)d\nu_i\right)\cdot\int_{u-\triangle-q_{k+1}(D)}^\infty p(\nu_{k+1})d\nu_{k+1}du$$

$$=\mathbb{E}_{z\sim p_\rho}\left[\left(\frac{p_\rho(z-\triangle)}{p_\rho(z)}\right)\left(\prod_{i\leq k}\int_{-\infty}^{z-\triangle-q_i(D)}p(\nu_i)d\nu_i\right)\cdot\int_{z-\triangle-q_{k+1}(D)}^\infty p(\nu_{k+1})d\nu_{k+1}\right]$$

where in the last line, we rename the variable $u$ back to $z$.

Substituting the above expression to the definition of RDP and apply Jensen's inequality

$$(6)=\sum_{k=0}^\infty\frac{\mathbb{E}_{z\sim p_\rho}\left[\left(\frac{p_\rho(z-\triangle)}{p_\rho(z)}\right)\left(\prod_{i\leq k}\int_{-\infty}^{z-\triangle-q_i(D)}p(\nu_i)d\nu_i\right)\cdot\int_{z-\triangle-q_{k+1}(D)}^\infty p(\nu_{k+1})d\nu_{k+1}\right]^\alpha}{\mathbb{E}_{z\sim p_\rho}\left[\left(\prod_{i\leq k}\int_{-\infty}^{z-q_i(D')}p(\nu_i)d\nu_i\right)\cdot\int_{z--q_{k+1}(D')}^\infty p(\nu_{k+1})d\nu_{k+1}\right]^{\alpha-1}}$$

$$\leq\sum_{k=0}^\infty\mathbb{E}_{z\sim p_\rho}\frac{\left(\frac{p_\rho(z-\triangle)}{p_\rho(z)}(\prod_{i\leq k}\int_{-\infty}^{z-\triangle-q_i(D)}p(\nu_i)d\nu_i)\int_{z-\triangle-q_{k+1}(D)}^\infty p(\nu_{k+1})d\nu_{k+1}\right)^\alpha}{\left((\prod_{i\leq k}\int_{-\infty}^{z-q_i(D')}p(\nu_i)d\nu_i)\int_{z-q_{k+1}(D')}^\infty p(\nu_{k+1})d\nu_{k+1}\right)^{\alpha-1}}\tag{7}$$

The inequality applies Jensen's inequality to bivariate function $f(x,y)=x^\alpha y^{1-\alpha}$, which is jointly convex on $\mathcal{R}_+^2$ for $\alpha\in(1,+\infty)$.

Exchange the order of integral variable $z$ and $k$ in (7), we get

$$(7)=\mathbb{E}_{z\sim p_\rho}\left[\left(\frac{p_\rho(z-\triangle)}{p_\rho(z)}\right)^\alpha\sum_{k=0}^\infty\frac{\left((\prod_{i\leq k}\int_{-\infty}^{z-\triangle-q_i(D)}p(\nu_i)d\nu_i)\int_{z-\triangle-q_{k+1}(D)}^\infty p(\nu_{k+1})d\nu_{k+1}\right)^\alpha}{\left((\prod_{i\leq k}\int_{-\infty}^{z-q_i(D')}p(\nu_i)d\nu_i)\int_{z-q_{k+1}(D')}^\infty p(\nu_{k+1})d\nu_{k+1}\right)^{\alpha-1}}\right]$$

$$=\mathbb{E}_{z\sim p_\rho}\left[\left(\frac{p_\rho(z-\triangle)}{p_\rho(z)}\right)^\alpha\sum_{k=0}^\infty\frac{\left(\prod_{i=1}^k\Pr[q_i(D)+\triangle+\nu_i<z|z]\Pr[q_{k+1}(D)+\triangle+\nu_{k+1}\geq z|z]\right)^\alpha}{\left(\Pr_{\mathcal{M}(D')}[K=k+1|z]\right)^{\alpha-1}}\right]\tag{8}$$

$$=\mathbb{E}_{z\sim p_\rho}\left[\left(\frac{p_\rho(z-\triangle)}{p_\rho(z)}\right)^\alpha\mathbb{E}_{K\sim\mathcal{M}(D')}\left[\frac{\left(\prod_{i=1}^{K-1}\Pr[\tilde{q}_i(D)+\nu_i<z|z]\Pr[\tilde{q}_K(D)+\nu_K\geq z|z]\right)^\alpha}{(\Pr[\mathcal{M}(D')=K|z])^\alpha}\middle|z\right]\right]$$

$$=\mathbb{E}_{z\sim p_\rho}\left[\left(\frac{p_\rho(z-\triangle)}{p_\rho(z)}\right)^\alpha\mathbb{E}_{K\sim\mathcal{M}(D')}\left[\frac{\left(\Pr[\mathcal{M}(D)=K|z,\tilde{\mathcal{Q}}]\right)^\alpha}{(\Pr[\mathcal{M}(D')=K|z])^\alpha}\middle|z\right]\right]\tag{9}$$

which completes the proof.

To understand the last step: recall our definition of fictitious query $\tilde{q}_i$, which obeys $\tilde{q}_i(D) = q_i(D) + \triangle$ and $\tilde{q}_i(\tilde{D}) = q_i(\tilde{D})$ for all other dataset $\tilde{D} \neq D$. Observe that the expression in the numerator of (8) actually describes a valid probability distribution of $K$, which says the probability of $\mathcal{M}(D)$ stopping at time $K = k + 1$ when the sequence of input is $\tilde{q}_1, ..., \tilde{q}_k + 1, ...,$ i.e.,

$$\prod_{i=1}^{k} \Pr[q_i(D) + \Delta + \nu_i < z | z] \Pr[q_{k+1}(D) + \Delta + \nu_i \geq z | z] = \Pr_{\mathcal{M}(D)}[K = k + 1 | z, \tilde{q}_1, \tilde{q}_2, ...].$$

The conditioning on the sequence of queries might appear to be new, but recall that all our probabilities are conditioned on a sequence of queries that are chosen from $\mathcal{Q}(\triangle)$ or $\mathcal{Q}_+(\triangle)$ to begin with. They are just not written out explicitly. This is an instance, where we actually need to condition on a different set of queries to formally write down this valid probability distribution above. □

**Remark.** The lemma de-convolves the moment of interests into the mixture of conditional moments of another two distributions that can be written down explicitly. The proof is delicate but informative, as it explicitly leveraging the fact that $\mathcal{M}_\nu$ is a *noise-adding* mechanism, so as to argue the implication of the randomization for a *different* query $\tilde{q}_i$ than the one that it seems to be intending for according to the algorithm $q_i$. There are several other novel components. We encourage readers to check it out in details.

A remarkable consequence of this lemma is that we can essentially cancel all factors concerning $\perp$s.

$$(*) = \mathbb{E}_{K \sim \mathcal{M}(D')} \left[ \frac{\left( \prod_{i=1}^{K-1} \Pr[q_i(D) + \triangle + \nu_i < z | z] \Pr[q_K(D) + \triangle + \nu_i \geq z | z] \right)^{\alpha}}{\left( \prod_{i=1}^{K-1} \Pr[q_i(D') + \nu_i < z | z] \Pr[q_K(D') + \nu_i \geq z | z] \right)^{\alpha}} \middle| z \right]$$

$$\leq \mathbb{E}_{K \sim \mathcal{M}(D')} \left[ \frac{(\Pr[q_K(D) + \triangle + \nu_i \geq z | z])^{\alpha}}{(\Pr[q_K(D') + \nu_i \geq z | z])^{\alpha}} \middle| z \right] \tag{10}$$

The inequality in the last line uses the fact that $q_i$ has a global sensitivity of $\triangle$, which implies that $\Pr[q_i(D) + \triangle + \nu_i < z | z] \leq \Pr[q_i(D') + \nu_i < z | z].$ for all $i$.

Further observe that $\tilde{q}_K$ has a sensitivity of $2\triangle$ since $q_K$ has sensitivity $\triangle$. By the property of the noise-adding mechanism $\mathcal{M}_\nu$, it obeys $\epsilon_\nu(\alpha)$-RDP for all queries having having sensitivity $2\triangle$. Therefore, if $\epsilon_\nu(\infty) < +\infty$, then we can bound (10) with $e^{\alpha \epsilon_\nu(\infty)}$. In fact, this bound can be improved slightly if we directly work with (5), which we state as a lemma.

**Lemma 18.** *If $\epsilon_\nu(\infty) < +\infty$, then the expression $(*)$ in (5) obeys $(*) \leq e^{(\alpha-1)\epsilon_\nu(\infty)}$.*

*Proof.* We use a trick due to [Bun and Steinke, 2016] with some modifications.

First, check that by our trivial bounds

$$0 \leq \frac{\Pr[\mathcal{M}(D) = K | z, \tilde{\mathcal{Q}}]}{\Pr[\mathcal{M}(D') = K | z]} \leq e^{\epsilon}$$

Define random function $A(K)$ supported on $\{0, e^{\epsilon}\}$ such that $\mathbb{E}[A(K)|K] = \frac{\Pr[\mathcal{M}(D)=K|z,\tilde{\mathcal{Q}}]}{\Pr[\mathcal{M}(D')=K|z]}$.

Note that when $\alpha = 1$

$$\Pr[A(K) = e^{\epsilon}] \cdot e^{\epsilon} = \mathbb{E}_K[\mathbb{E}[A(K)|K]] = \mathbb{E}_{K \sim \mathcal{M}(D')} \left[ \frac{\Pr[\mathcal{M}(D) = K | z, \tilde{\mathcal{Q}}]}{\Pr[\mathcal{M}(D') = K | z]} \right] = 1.$$

The first moment is equal to 1 critically relies on our construction where the numerator in the expectation of $(*)$ is the $\alpha$th power of a valid probability distributions.

This implies that $\Pr[A(K) = e^{\epsilon}] = e^{-\epsilon}$, therefore

$$(*) = \mathbb{E}_K[\mathbb{E}[A(K)|K]^{\alpha}] \overset{\underset{\mathrm{Jensen}}{\downarrow}}{\leq} \mathbb{E}[\mathbb{E}[A(K)^{\alpha}|K]] = \mathbb{E}[A(K)^{\alpha}] = \Pr[A(K) = e^{\epsilon}] \cdot e^{\alpha\epsilon} = e^{(\alpha-1)\epsilon},$$

which completes the proof. □

Now we are ready to prove the three claims of Theorem 8.

**The claim** (3): Substitute the the above bound into Lemma 17, we get:

$$\mathbb{E}_{o\sim\mathcal{M}(D')}\left[\left(\frac{\Pr[\mathcal{M}(D)=o]}{\Pr[\mathcal{M}(D')=o]}\right)^\alpha\right]$$

$$\overset{\substack{\text{Lemma 17 and 18}\\\downarrow}}{\leq}\mathbb{E}_{z\sim p_\rho}\left[\left(\frac{p_\rho(z-\triangle)}{p_\rho(z)}\right)^\alpha\right]e^{(\alpha-1)\epsilon_\nu(\infty)}\leq e^{(\alpha-1)\epsilon_\rho(\alpha)}e^{(\alpha-1)\epsilon_\nu(\infty)}.$$

where the second inequality in the last line uses the definition of RDP of $\mathcal{M}_\rho$, for a trivial query $q(D)=-\triangle, q(D')=0$. (3) follows by simply taking $\log(\cdot)/(\alpha-1)$ on both sides.

**The claim** (1) **and** (2): To get the other two bounds, we need an alternative analysis of $(**)$. To avoid crowded notations, we drop the conditioning on $z$ from $\Pr[\cdot|\rho=z]$. By the definition of expectation,

$$(*)\leq(10)=\sum_{k=0}^\infty\prod_{i=0}^k\Pr[q_i(D')+\nu_i<z]\Pr[q_{k+1}(D')+\nu_{k+1}\geq z]\frac{\Pr[\tilde{q}_{k+1}(D)+\nu_{k+1}\geq z]^\alpha}{\Pr[\tilde{q}_{k+1}(D')+\nu_{k+1}\geq z]^\alpha}$$

$$\overset{\substack{\tilde{q}=q\text{ on }D'\\\downarrow}}{=}\sum_{k=0}^\infty\prod_{i=0}^k\Pr[q_i(D')+\nu_i<z]\frac{\Pr[\tilde{q}_{k+1}(D)+\nu_{k+1}\geq z]^\alpha}{\Pr[\tilde{q}_{k+1}(D')+\nu_{k+1}\geq z]^{\alpha-1}}$$

$$\overset{\substack{\text{Lemma 15}\\\downarrow}}{\leq}\left(\sum_{k=0}^\infty\prod_{i=0}^k\Pr[\tilde{q}_i(D')+\nu_i<z]\right)\cdot e^{\epsilon(\alpha)(\alpha-1)}. \tag{11}$$

In the last line, we applied the "indistinguishability" property of an RDP mechanism in Lemma 15 for the particular event $S=x\in\mathbb{R}|x\geq z$, for the random-variable $\mathcal{M}(D,\tilde{q}_{k+1})$ and $\mathcal{M}(D',\tilde{q}_{k+1})$ in the numerator and denominator respectively.

The issue is how to proceed. $\sum_{k=0}^\infty\prod_{i=0}^k\Pr[\tilde{q}_i(D')+\nu_i<z]$ does not sum to 1 because $\prod_{i=0}^k\Pr[\tilde{q}_i(D')+\nu_i<z]$ is not a probability distribution of $k$. The saving grace is the following alternative definition of expectation.

**Lemma 19.** *For a non-negative random variable $X$, $\mathbb{E}[X]=\int_0^\infty\Pr[X>x]dx$.*

Recall that $K$ is the first index of $\top$, we can rewrite $\prod_{i=0}^k\Pr[\tilde{q}_i(D')+\nu_i<z]$ as $\Pr_{D'}[K>k|z]$. Thus

$$\sum_{k=0}^\infty\prod_{i=0}^k\Pr[\tilde{q}_i(D')+\nu_i<z]=\sum_{k=0}^\infty\Pr_{D'}[K>k|z]=\mathbb{E}[K|z] \tag{12}$$

It follows that

$$\mathbb{E}_{o\sim\mathcal{M}(D')}\left[\left(\frac{\Pr[\mathcal{M}(D)=o]}{\Pr[\mathcal{M}(D')=o]}\right)^\alpha\right]\leq\mathbb{E}_{z\sim p_\rho}\left[\left(\frac{p_\rho(z-\triangle)}{p_\rho(z)}\right)^\alpha\mathbb{E}[K|z]\right]e^{\epsilon_\nu(\alpha)(\alpha-1)}$$

Claim (1) uses $\mathbb{E}[K|z]\leq k_{\max}+1$. By using a different Holder's inequality with conjugate pair $\gamma$ and $\gamma*:=\gamma/(\gamma-1)$, we obtain

$$\mathbb{E}_{o\sim\mathcal{M}(D')}\left[\left(\frac{\Pr[\mathcal{M}(D)=o]}{\Pr[\mathcal{M}(D')=o]}\right)^\alpha\right]\leq\mathbb{E}_{z\sim p_\rho}\left[\left(\frac{p_\rho(z-\triangle)}{p_\rho(z)}\right)^{\gamma^*\alpha}\right]^{1/\gamma^*}\cdot\left(\mathbb{E}_{z\sim p_\rho}\left[\mathbb{E}[K|z]^\gamma\right]\right)^{1/\gamma}\cdot e^{\epsilon_\nu(\alpha)(\alpha-1)}$$

(2) follows by taking $\log(\cdot)/(\alpha-1)$ on both sides and applying the definition of RDP. This completes the proof of Theorem 8.

## C  Other proofs of technical results

**Proposition 20** (Restatement of Proposition 10 with mroe details)**.** Let Algorithm 2 be instantiated with $\mathcal{Q}_+(\triangle)$, $\mathcal{M}_\rho$ and $\mathcal{M}_\nu$ be Gaussian mechanism with parameter $\sigma_1$ and $\sigma_2$. Then for all $T<+\infty$ and $\gamma>1$ such that $\sigma_2>\sqrt{\gamma}\sigma_1$, Algorithm 2 with $c=1$ halts with $K$ rounds satisfying

$$\mathbb{E}_\rho[\mathbb{E}[K|\rho=z]^\gamma]\leq\int_{-\infty}^\infty\frac{1}{\sigma_1}\phi(z/\sigma_1)\left(\frac{\Phi((T+z)/\sigma_2)}{1-\Phi((T+z)/\sigma_2)}\right)^\gamma dz<+\infty, \tag{13}$$

where $\phi(x) = \frac{e^{-x^2}}{\sqrt{2\pi}}$ and $\Phi(x) = \int_{-\infty}^{x} \phi(x)dx$ are the pdf and CDF of the standard normal distribution. If $\sigma_2 \geq \sqrt{\gamma + 1}\sigma_1$, then a more interpretable bound of the above is

$$\mathbb{E}[\mathbb{E}[K|\rho = z]^\gamma] \leq 1 + (c_\gamma \sqrt{2\pi} \max\{\frac{T(1+\gamma)}{\sigma_1}, 1\})^\gamma (1+\gamma)^{1/2} e^{\frac{\gamma T^2}{2\sigma_1^2}}$$

where $c_\gamma$ is a universal constant that comes from the moments bounds and depends only on $\gamma$. For the special case when $\gamma = 2$, and $\sigma_2 = \sqrt{3}\sigma_1$, we get $\mathbb{E}[\mathbb{E}[K|\rho = z]^2] \leq 1 + 2\sqrt{3}\pi(1 + \frac{9T^2}{\sigma_1^2})e^{\frac{T^2}{\sigma_1^2}}$.

*Proof of Proposition 20.* Consider the case when all queries are non-negative, and the threshold $T$ is given, then the datasets that maximizes all moments of $K|\rho = z$ for all $z$ are given by $f_i(D) = 0$ for all $i$. Notice that $K|\rho = z$ follows a Negative Binomial Distribution, thus

$$\mathbb{E}[K|z] = \frac{F_v[T+z]}{1 - F_v[T+z]},$$

where $F_v$ is the cumulative density function (CDF) of the noise $v$. The moments of $\mathbb{E}[K|z]$, when exists, can be computed by numerical integration. When $z \sim \mathcal{N}(0, \sigma_1^2)$ and $v \sim \mathcal{N}(0, \sigma_2^2)$ for $\sigma_2 > \sigma_1\sqrt{\gamma}$, we can work out bounds of the $\gamma$th moments of $\mathbb{E}[K|z]$.

Let $\phi$ be the standard normal density function and $\Phi$ be the CDF. There is a lower bound of the Gaussian tail for all $x > 0$

$$1 - \Phi(x) \geq \frac{x}{x^2 + 1}\phi(x)$$

Thus for $y \geq -T + \sigma_2$, we have

$$\mathbb{E}[\mathbb{E}[K|z]^\gamma] = \int_{-\infty}^{\infty} \frac{1}{\sigma_1}\phi(z/\sigma_1)\left(\frac{\Phi((T+z)/\sigma_2)}{1 - \Phi((T+z)/\sigma_2)}\right)^\gamma dz$$

$$= \int_{-\infty}^{y} \frac{1}{\sigma_1}\phi(z/\sigma_1)\left(\frac{\Phi((T+z)/\sigma_2)}{1 - \Phi((T+z)/\sigma_2)}\right)^\gamma dz + \int_{y}^{\infty} \frac{1}{\sigma_1}\phi(z/\sigma_1)\left(\frac{\Phi((T+z)/\sigma_2)}{1 - \Phi((T+z)/\sigma_2)}\right)^\gamma dz$$

$$\leq \int_{-\infty}^{y} \frac{1}{\sigma_1}\phi(z/\sigma_1)dz + \int_{y}^{\infty} \frac{1}{\sigma_1}\phi(z/\sigma_1)\left(\frac{\frac{(T+z)^2}{\sigma_2^2} + 1}{\frac{(T+z)}{\sigma_2}\phi((T+z)/\sigma_2)}\right)^\gamma dz$$

$$\overset{\underset{T+y \geq \sigma_2}{\downarrow}}{\leq} \Phi(y/\sigma_1) + \int_{y}^{\infty} \frac{1}{\sqrt{2\pi}\sigma_1}e^{-\frac{z^2}{2\sigma_1^2}}(2\pi)^{\gamma/2}e^{\frac{\gamma(T+z)^2}{2\sigma_2^2}}2^\gamma\frac{(T+z)^\gamma}{\sigma_2^\gamma}dz$$

$$\overset{\underset{u:=z+T, \tilde{\sigma}:=(\frac{1}{\sigma_1^2} - \frac{\gamma}{\sigma_2^2})^{-1/2}}{\downarrow}}{=} \Phi(y/\sigma_1) + (2\pi)^{\frac{\gamma}{2}}\frac{\tilde{\sigma}}{\sigma_1}\int_{T+y}^{\infty}\frac{u^\gamma}{\sigma_2^\gamma}\frac{1}{\sqrt{2\pi}\tilde{\sigma}}e^{-\frac{u^2}{2\tilde{\sigma}^2} + \frac{2uT}{2\sigma_1^2} - \frac{T^2}{2\sigma_1^2}}du$$

$$= \Phi(y/\sigma_1) + (2\pi)^{\frac{\gamma}{2}}\frac{\tilde{\sigma}}{\sigma_1}e^{-\frac{T^2}{2\sigma_1^2} + \frac{T^2\tilde{\sigma}^2}{2\sigma_1^4}}\int_{T+y}^{\infty}\frac{u^\gamma}{\sigma_2^\gamma}\frac{1}{\sqrt{2\pi}\tilde{\sigma}}e^{-\frac{(u-T\frac{\tilde{\sigma}^2}{\sigma_1^2})^2}{2\tilde{\sigma}^2}}du$$

$$\overset{\underset{\text{take } |\cdot| \text{ and relax the range of integral}}{\downarrow}}{\leq} \Phi(y/\sigma_1) + (2\pi)^{\frac{\gamma}{2}}\frac{\tilde{\sigma}}{\sigma_1}e^{\frac{\gamma T^2}{2(\sigma_2^2 - \gamma\sigma_1^2)}}\mathbb{E}_{X \sim \mathcal{N}(\frac{T\tilde{\sigma}}{\sigma_1^2}, 1)}[|X|^\gamma]$$

where $\mathbb{E}_{X \sim \mathcal{N}(\frac{T\tilde{\sigma}}{\sigma_1^2}, 1)}[|X|^\gamma]$ is the $m$th non-central moments which is on the order of $\max\{\frac{T\tilde{\sigma}}{\sigma_1^2}, 1\}^\gamma$ — and can be evaluated in a closed-form. Finally, we can simply take $y - T + \sigma_2$.

Now, suppose we take $\sigma_2 = \sqrt{1+\gamma}\sigma_1$, then we get $\tilde{\sigma} = \sigma_2$. The above bound simplifies to:

$$\mathbb{E}[\mathbb{E}[K|z]^\gamma] \leq 1 + (c_\gamma\sqrt{2\pi}\max\{\frac{T(1+\gamma)}{\sigma_1}, 1\})^\gamma(1+\gamma)^{1/2}e^{\frac{\gamma T^2}{2\sigma_1^2}}$$

where $c_\gamma$ is a universal constant that comes from the moments bounds and depends only on $\gamma$. If $\gamma = 2$ and $\sigma_2 = \sqrt{3}\sigma_1$, then $\tilde{\sigma} = \sigma_2$, and

$$\mathbb{E}[\mathbb{E}[K|z]^2] \leq 1 + 2\sqrt{3}\pi(1 + \frac{9T^2}{\sigma_1^2})e^{\frac{T^2}{\sigma_1^2}}.$$

$\square$

## C.1 RDP analysis with $c \geq 1$, proof of Theorem 11

**Theorem 21** (Restatement of Theorem 11, RDP for length-capped SVT with $c > 1$). *The generalized SVT with cut-off parameter $c > 1$ and a maximum length is $k_{\max}$ obeys that*

$$\mathbb{D}_\alpha(\mathcal{M}(D)\|\mathcal{M}(D')) \leq \epsilon_\rho(\alpha) + c\epsilon_\nu(\alpha) + \frac{\log \sum_{k=0}^{c} \binom{k_{\max}}{k}}{\alpha - 1}.$$

*After a careful revision, we found there is a minor typo in the statement of Theorem 11 (the $1/(\alpha-1)$ term shouldn't be there) we provide the correct version above.

The proof follows a similar sequence of arguments to that we presented for $c = 1$.

When $c > 1$, the output space of the algorithm is $S = \{\top, \bot\}^\ell, \ell = 0, 1, ..., k_{\max}$ with the additional restriction that the number of $\top$s are smaller than $c$. Denote $I_\bot := \{i : o_i = \bot\}$ and $I_\top := \{j : o_j = \top\}$. Then we can write the probability of outputting $o$ as following:

$$\Pr[\mathcal{M}(D) = o] = \int_{-\infty}^{+\infty} p_\rho(z) \left( \prod_{i \in I_\bot} \int_{-\infty}^{z-q_i(D)} p(\nu_i)d\nu_i \right) \left( \prod_{j \in I_\top} \int_{z+q_j(D)}^{\infty} p(\nu_j)d\nu_j \right) dz$$

Similarly, we have

$$\mathbb{E}_o \left[ \left( \frac{\Pr[\mathcal{M}(D) = o]}{\Pr[\mathcal{M}(D') = o]} \right)^\alpha \right] = \sum_{o \in S} \Pr[\mathcal{M}(D') = o] \left( \frac{\Pr[\mathcal{M}(D) = o]}{\Pr[\mathcal{M}(D') = o]} \right)^\alpha \qquad (*)$$

Apply the same logic from the proof for $c = 1$, we can upper bound $\Pr[\mathcal{M}(D) = o]$ in the following

$$\Pr[\mathcal{M}(D) = o] \leq \mathbb{E}_{z \sim p_\rho} \left( \frac{p_\rho(z - \triangle)}{p_\rho(z)} \prod_{i \in I_\bot} \int_{-\infty}^{z-q_i(D')} p(\nu_i)d\nu_i \right) \left( \prod_{j \in I_\top} \int_{z+\triangle+q_j(D)}^{\infty} p(\nu_j)d\nu_j \right) \tag{14}$$

Then apply Jensen' inequality to $\frac{\Pr[\mathcal{M}(D)=o]^\alpha}{\Pr[\mathcal{M}(D')=o]^{\alpha-1}}$, we have

$$(14) \leq \sum_{o \in S} \mathbb{E}_{z \sim p_\rho} \frac{\left( \frac{p_\rho(z-\triangle)}{p_\rho(z)} \prod_{i \in I_\bot} \int_{-\infty}^{z-q_i(D')} p(\nu_i)d\nu_i \right)^\alpha \left( \prod_{j \in I_\top} \int_{z+\triangle+q_j(D)}^{\infty} p(\nu_j)d\nu_j \right)^\alpha}{\left( \prod_{i \in I_\bot} \int_{-\infty}^{z-q_i(D')} p(\nu_i)d\nu_i \right)^{\alpha-1} \left( \prod_{j \in I_\top} \int_{z+q_j(D')}^{\infty} p(\nu_j)d\nu_j \right)^{\alpha-1}} \tag{15}$$

Exchange the order of integral in $z$ and $o$, we get

$$(15) = \underbrace{\mathbb{E}_{z \sim p_\rho} \left( \frac{p_\rho(z - \triangle)}{p_\rho(z)} \right)^\alpha \sum_{o \in S} \Pr[\prod_{i \in I_\bot} q_i(D') + \nu_i < z]}_{\text{denote by}(**)} \cdot \frac{\left( \prod_{j \in I_\top} \Pr[q_j(D) + \nu_j + \triangle \geq z] \right)^\alpha}{\left( \prod_{j \in I_\top} \Pr[q_j(D') + \nu_j \geq z] \right)^{\alpha-1}}$$

In the case of length-capped SVT, the algorithm stops whenever $|o| \geq k_{\max}$ or $|I_\top| \geq c$. By the fact that probabilities $\leq 1$, we use the following crude bound

$$\sum_{o \in S} \Pr[\prod_{i \in I_\bot} q_i(D') + \nu_i < z] \leq \sum_{o \in S} 1 = |S|,$$

i.e., the cardinality of the output space, which is bounded from above by $\sum_{k=0}^{c} \binom{k_{\max}}{k}$.

Moreover, we can bound the $\mathbb{E}_{z \sim p_\rho} \left( \frac{p_\rho(z-\triangle)}{p_\rho(z)} \right)^\alpha$ term with $e^{(\alpha-1)\epsilon_\rho(\alpha)}$ using the definition of RDP (with a trivial query that outputs 0 and $\triangle$ for $D$ and $D'$ as we constructed before). Therefore, $(**)$ is bounded by $e^{(\alpha-1)\epsilon_\rho(\alpha)} \cdot \sum_{k=0}^{c} \binom{k_{\max}}{k}$.

For the second part $\frac{(\prod_{j \in I_\top} \Pr[q_j(D) + \nu_j + \triangle \geq z])^\alpha}{(\prod_{j \in I_\top} \Pr[q_j(D') + \nu_j \geq z])^{\alpha-1}}$, we apply the same trick of defining a sequence of fictitious queries $\tilde{q}_1, ..., \tilde{q}_{k_{\max}}$ as in 4. For each $j \in I_\top$, $\frac{\Pr[\tilde{q}_j(D) + \nu_j + \triangle \geq z]}{\Pr[\tilde{q}_j(D') + \nu_j \geq z])^{\alpha-1}} \leq e^{\epsilon_\nu(\alpha)(\alpha-1)}$ using the "indistinguishability" property of an RDP mechanism defined in Lemma 15. Since $|I_\top| \leq c$, the second part is bounded by $e^{c\epsilon_\nu(\alpha)(\alpha-1)}$.

## D   Different ways of doing composition

There are several different ways of achieving strong composition for SVT.

### D.1   RDP Composition of generalized SVT with $c = 1$

In the case when a finite $k_{\max}$ is enforced, SVT is able to process up to $ck_{\max}$ queries, and has a total RDP bound of $c$ times

$$\epsilon(\alpha) \leq c\epsilon_\rho(\alpha) + c\epsilon_\nu(\alpha) + \frac{c\log(1 + k_{\max})}{\alpha - 1}.$$

**Corollary 22** (($\epsilon, \delta$)-DP of Gaussian SVT with $c \geq 1$ with RDP composition)**.** When both noise are Gaussian we get an overall ($\epsilon, \delta$)-DP with

$$\epsilon(\delta) \leq \frac{c\Delta^2}{2\sigma_1^2} + \frac{2c\Delta^2}{\sigma_2^2} + 2\sqrt{c(\frac{\Delta^2}{2\sigma_1^2} + \frac{2\Delta^2}{\sigma_2^2})(\log(\delta^{-1}) + c\log((1 + k_{\max})))}$$

The results recover the strong composition that gives an $(O(\frac{\sqrt{c\log(1/\delta)}\triangle}{\sigma_1}), \delta)$-DP guarantee up to $c = O(\sqrt{\sigma_1^2/\triangle^2})$, provided that $\delta \leq (k_{\max})^{-c}$. In the case of nonnegative query case, we may obtain a similar bound for each valid $\gamma$ which requires only $\delta \leq (O(\frac{T^2}{\sigma_1^2}))^{-c}$. For larger $c$ or large $\delta$, the privacy losses increase linearly with $c$.

### D.2   Directly use the RDP of generalized SVT with $c > 1$

We discussed this in the remark underneath Theorem 11, which says that we obtain ($\epsilon, \delta$)-DP with

$$\epsilon(\delta) \leq \frac{\Delta^2}{2\sigma_1^2} + \frac{2c\Delta^2}{\sigma_2^2} + \sqrt{2\left(\frac{\Delta^2}{2\sigma_1^2} + \frac{2c\Delta^2}{\sigma_2^2}\right)\left(\log(\delta^{-1}) + \log c\binom{k_{\max}}{c}\right)}$$

The story is similar to the case for $c = 1$, we get the $O(\sqrt{c})$-type strong composition when $\delta \propto k_{\max}^{-c}$, and $O(c)$-type composition when $\delta < k_{\max}^{-1}$.

### D.3   Hybrid composition based on both RDP and KOV-composition

The restriction on small $\delta$ is quite limiting, which motivated us to consider the hybrid scheme for composition as in the Stagewise generalized SVT (Algorithm 3) which chooses $c'$ according to the pre-specified $\delta$. This essentially allows us to replace $c$ with $\log(c/\delta)$ as in Theorem 12.

## E   Connections of Noisy Screening and its Data-Dependent Privacy Bound

NoisyScreening and its data-dependent privacy loss computation is an alternative way of viewing the problem that SVT addresses. Similar to SVT, given a threshold, noisy screening output $\perp$ if $q_t(D) + \mathcal{N}(0, \sigma^2) \geq T$ and $\top$ otherwise. Notice that this is a fixed algorithm that obeys $(\alpha, \frac{\alpha k \triangle^2}{\sigma^2})$-RDP when running for $k$ iterations by the post-processing property and composition of Gaussian mechanisms. However, when $|q_t(D) - T|$ is large, then the post-processing of the output of Gaussian mechanism actually amplifies the privacy guarantee due to the overwhelming probability of outputting either $\perp$ or $\top$ on $D$ and all neighboring dataset $D'$ to $D$, (see details in [Papernot et al., 2018, Theorem 6]).

The data-dependent analysis of Noisy screening behaves like SVT as it pays a privacy loss only when the query $q_t(D)$ is close to the threshold $T$. The notion of privacy loss that is achieved by noisy-screening has a similar notion of DP but it is data-dependent therefore the value itself is considered private information. Part of our motivation in this work is to investigate whether we can obtain formally differentially private methods that replicates the practical performance of data-dependent privacy analysis for noisy screening while satisfying a pre-specified privacy budget.

# F  Applications

## F.1  Adaptive data analysis

The Private-Guess-and-Check algorithm is an application of sparse vector technique that assumes that the analyst sends a "guess" with each query. The curator will then check if the guess is sufficiently accurate by SVT and return the "guess" without tapping into the data and move to the next round and only release something from the data using, e.g., Gaussian mechanism if the "guess" fails to be accurate. In this way, SVT will allow us to answer exponentially more queries while paying a privacy loss that essentially depends only on the number of "incorrect" guesses.

**Example 23.** *"Reuseable-Holdout" is one way to generate these guesses in the context of adaptive data analysis. In the original version, the algorithm split the dataset into $c+1$ folds and use each fold to answer only one "bad" query. A better version splits the dataset into just two, and then give one of the fold to the analyst for exploratory data analysis and for coming up with queries and guesses, then answer those queries using private-guess-and-check.*

**Example 24.** *"Private-Multiplicative-weight" [Hardt and Rothblum, 2010] is a classical DP algorithm that can be thought of as another instantiation of the Private-Guess-and-Check. It aims at iteratively refine a vector of weights on all possible datasets (each one of them is an expert) using the Hedge algorithm from online learning to produce "guesses" in the form of an exponentially weighted averages. The algorithm using SVT such that it only releases the private answers to the queries if the answer is sufficiently different from the "guess". The regret bound of the Hedge algorithm yields a bound of $c$ that implies a bound on the wrong "guesses", which then allows answering exponentially many queries accurately.*

**Lemma 25** (Theorem 3.5 [Jung et al., 2020] ). *Suppose $\mathcal{M}$ is $(\epsilon, \delta)$-differentially private and $(\alpha, \beta)$-accurate with respect to samples. Then for any analyst $\mathcal{A}$ that chooses a sequence of queries adaptively, we have that*

$$\Pr_{\substack{Data \sim \mathcal{D} \\ (\hat{f}_{\mathcal{I}}, \mathcal{I}) \sim Interaction(\mathcal{A}, \mathcal{M}|Data)}} \left[ \max_{i \in \mathcal{I}} \left| \hat{f}_i - \mathbb{E}_{Data' \sim \mathcal{D}} \left[ f_i(Data') \right] \right| \geq \alpha + (e^\epsilon - 1) + v_1 + 2v_2 \right] \leq \frac{\beta}{v_1} + \frac{\delta}{v_2}.$$

*Proof of Theorem 14.* Let the threshold that we use $T = \frac{\sigma \sqrt{2 \log(k/\delta)}}{n}$, then it is clear that the answers are $\left( \sigma \sqrt{2 \log(k/\delta)}, \delta \right)$-sample-accurate using the concentration bound of Gaussian noise. Now by the composition of $c$-Gaussian mechanisms and the Gaussian-SVT with parameter $c = 1, k, \sigma, \triangle = 1/n$, the whole procedure is $\left( O(\sqrt{\frac{c}{n^2 \sigma^2} \log(1/\delta)^2}), \delta \right)$-DP. By choosing $\sigma \asymp c^{1/4} n^{-1/2} \log(k/\delta)^{1/4}$, and then pick $v_1 = v_2 = O(c^{1/4} n^{-1/2} \log(k/\delta)^{3/4})$, then we obtain the high-probability generalization bound for adaptive queries as claimed. $\square$

**Remark.** In comparison, if we use Laplace mechanism-based SVT for the same problem, the answers will only be $(\log(k/\delta)/(n\epsilon), \delta)$-sample-accurate, due to the perturbation of the threshold. It will be $(O(\epsilon \sqrt{c \log(1/\delta)}), \delta)$-DP. By choosing $\epsilon$ appropriately to balance the two terms, we get that the standard SVT also achieves the same bound of $O(c^{1/4} \log(k/\delta)^{3/4} n^{-1/2})$.

## F.2  Model-agnostic private learning

Model-agnostic private learning is another cute application of the sparse-vector technique. In this problem, the learner has access to a private labeled dataset and a public unlabeled dataset. The algorithm leverages a blackbox learner, e.g., a deep learning algorithm, by training one classifier on each randomly split of the private dataset. Then it privately labels the public dataset by privately releasing the majority-votes of these classifiers' predictions.

This scheme has been shown to be practical [Papernot et al., 2017, 2018] by combining simple Gaussian mechanism for differential privacy with semi-supervised learning approaches. Bassily et al. [2018] substantially improves the algorithm by showing that, under a PAC-learning framework, that one can privately release the labels for all public data points, while spending the privacy budget only for those data points where the voters are labeling incorrectly (or labeling inconsistently, to be more general).

SVT is applied to test whether each query has received an overwhelming majority from the voters by testing if distance-to-stability is sufficiently large. If so, the exact answer $f(D)$ is released with $\perp$ and if not, only $\top$ is released. Interestingly, this approach has a privacy loss that depends only on the number of $\top$s and it can be thought of as a composition of the SVT with the $(0, \delta)$-DP part of the event from the "Stability"-based argument.

While this approach provides a substantial benefit in theory, it has been observed in practice that it is often outperformed by simple Gaussian mechanism in practice, since the latter uses a more-concentrated noise and also a much tighter composition.

In the experiment section, we demonstrate that the story is now different when Gaussian SVT is used as a drop-in replacement.

## G  More details about the experiments

### G.1  Calibrating noise to privacy

In this section, given a predetermined privacy budget $(\epsilon, \delta)$ and the cut-off $c$, we evaluate SVT variants by comparing how many queries each SVT algorithm. Suppose we have an infinite sequence of queries with ground truth at 0 (null-hypothesis) and a fixed margin/ threshold. For each of the SVT-like algorithm, we calibrate the noise (or the length for Gaussian-based SVT) according to the privacy budget.

We estimate the length of answered queries with Negative Binomial Distribution. For example, in the case of Gaussian-SVT, when $z \sim \mathcal{N}(0, \sigma_1^2)$ and $\nu \sim \mathcal{N}(0, \sigma_2^2)$, denote $K$ as the number of queries answered when hits $c$. Notice that $K|\rho = z$ follows a Negative Binomial Distribution, $\mathbb{E}[K|z]$ can be estimated with $\frac{cF_\nu[T+z]}{1-F_\nu[T+z]}$, where $F_\nu$ is the CDF of the noise $\nu$ and queries are all zeros. Then by law of expectation, we can estimate $\mathbb{E}[K]$ with

$$\frac{1}{N} \sum_{i=1}^{N} \frac{c \cdot F_\nu[T + z_i]}{1 - F_\nu[T + z_i]}$$

$N$ is the number of trails and we sample $z_i \sim \mathcal{N}(0, \sigma_1^2)$ in each trail. We set $N = 10^5$ in our experiment.

We present our results in Figure 1, where the predefined privacy budget $\epsilon$ is varied at the $x$-axis. Regarding the choice of $k_{max}$, noting setting $k_{max}$ too large would lead the algorithm to hit $\top$ far before it reaches $k_{max}$ limit. Hence we set $k_{max}$ individually for each algorithm with each $\epsilon$ budget. For example, we set $k_{max} \approx 50 \cdot 2^{4\epsilon}$ for Gaussian-SVT with $T = 100$ and $k_{max} \approx 20 \cdot 10^{5\epsilon}$ with $T = 700$ such that $k_{max} \approx 1/(1 - F_\nu[T + z_i])$. The curves are unsmooth for some algorithms (e.g., Gaussian-SVT), since they would start a new subroutine when $k_{max}$ is achieved even before hitting $\top$. The left part is the low margin regime ($T = 100$) and $\delta$ is set to be $10^{-6}$. The purple line is the stage-wise generalized SVT, which can answer the largest number of queries across two regimes, especially in the high-margin regime. This is expected since when the margin $T$ is sufficiently high, the false positives will fall into the area of tails bound in either Laplace or Gaussian distribution. As discussed earlier, the Laplace distribution used in the SVT is heavy-tailed distribution, which would trigger the $c$ false positives sooner.

### G.2  Privacy cost for answering a full sequence

In this section, we evaluate SVT variants by comparing the composed privacy loss for finishing a fixed length sequence of queries.

Given a fixed sequence of $|Q| = 10^5$ queries with ground truth at 0, we fix the margin with $T = 1000$ for all algorithms, and align the standard deviation of Laplace noise and Gaussian noise, which is used

(a) Exp 2: Synthetic null sequence with aligned vari- (b) Exp 2: Synthetic null sequence with aligned tail
ance of noise                                            bound

Figure 3: Total composed privacy loss a the algorithm progressed for $\delta = 10^{-6}$. On the left, we fix the margin $T = 1000$ and $\sigma_1 = 210$ and align the variance of noise to perturb the queries. On the right, we align the tail bound for all algorithms by varying the noise scale with a fixed threshold $T = 1500$.

to perturb the query, i.e., $\sigma_2 = \sqrt{2}\lambda_2 = 240$. In Figure 3(a), the black-dash line describes the privacy cost of the Gaussian mechanism, serving as our baseline. Notably, with the same level noise to perturb queries, the Gaussian Mechanism's global sensitivity is the half of that in Gaussian-SVT. The brown line reports the privacy cost of data-dependent noisy screening before applying smooth-sensitivity analysis, which itself is considered private information. When the #iteration is small, all SVT-based algorithms have a flat region due to the number of $\top$ is zero.

We now evaluate SVT variants with an aligned tail bound. As shown in Figure 3(b), we fix the margin $T = 1500$ and vary $\gamma$ at the x-axis. For each $\gamma$, we choose the noise $\nu$ for each algorithm adaptively, such that the false positive rate is $\gamma$. For example, In the case of all Gaussian-based algorithms, the $\sigma_2$ is set such that $F_\nu[T] = 1 - \gamma$, where $\nu$ is drawn from $\mathcal{N}(0, \sigma_2^2)$ and $F_\nu$ is the CDF of the noise $\nu$. We see that Gaussian-SVT and Laplace-SVT perform similarly as their tail bounds are aligned and the stage-wise generalized SVT has the least privacy cost as it enjoys a sharper composition. The big gap between SVT-based algorithms and the Noisy Screening is due to the numerical issues in the calculation of the survival function: as $\gamma$ is small, e.g., $\gamma < 10^{-6}$, the scipy-based calculation in python will output 0 for $\Pr[q_t(D) + \mathcal{N}(0, \sigma_2^2) \geq T]$, which implied no false positive is detected. However, for all SVT-based algorithms, we enforce the false-positive $c$ to be at least 1, which can potentially yield a significant gap when $\gamma$ is small.

**Aligned variance or aligned FPR** Note that there are two undetermined parameters $(\sigma_2, \lambda_2)$ for SVT variants if we compare the privacy cost of answering a full sequence with a fixed margin of $T$. To build a meaningful connection between Laplace variants and Gaussian variants, we consider two types of alignments — variance of noise and FPR. There are other choices of alignments (e.g., mean, the third moment). Our consideration of the choice is to investigate the regions where Gaussian variants are advantageous. More specifically, with an aligned variance, we can observe the advantage of Gaussian SVT due to a thinner tail bound while this advantage disappears when we align the tail bound (FPR).

### G.3 Evaluation with real life data

In the application of model agnostic learning, Noisy screening is applied to test whether each query has received an overwhelming majority from the voters. We instantiate the task with the Private-kNN framework Zhu et al. [2020] using the CIFAR-10 dataset. The training set of CIFAR-10 is simulated as the private domain, and a sequence of queries are drawn from the public domain (testing set); for each query, we pick the top $K = 300$ closest neighbors from the private domain and output $\top$ if the plurality of neighbors is above a predetermined threshold.

In the privacy-preserving screening, we set the threshold $T = 210$ and fix the Gaussian noise $\sigma_2 = 80$ to perturb query for each algorithm. For Gaussian-SVT, we set $\sigma_1 = 40$ to perturb the threshold. The error bars are computed based on 10 independent run and has a correct 95% coverage.

### G.4 Adaptive data analysis

"Reuseable-Holdout" is one application of SVT in the context of adaptive data analysis. We instantiate the task using the experimental setups from Dwork et al. [2015a]. In this experiment, the analyst is given a $d$-dimensional labeled data $S$ of size $2n$, where each attribute is drawn independently from the normal distribution $\mathcal{N}(0, 1)$. The analyst splits $S$ randomly into a training set $S_t$ and a holdout set $S_h$ of equal size. The labels $y \in \{-1, 1\}$ are generated uniformly at random, so the data point and its label are not correlated. The goal of the analyst is to select variables to be included in the classifier. The analyst picks $k$ variables ($k < d$) with the label's largest absolute correlations with the training set. Then she verifies the correlations on the holdout set and selects only those variables whose correlation agrees in sign with the correlation on the training set, and the correlations are greater than a predefined threshold $T$. Then the analyst generates a linear threshold classifier using the selected variables and tests it on the holdout set. Full details can be found in the supplementary materials of Dwork et al. [2015a]. We set $n = 10000$, $d = 10000$ and varied the number of selected variables $k$. After a careful inspection of their code, we find that they use the Gaussian mechanism to select variables, rather than the Laplace-based SVT that they analyze. Note that this is also one motivation of this work — to find SVTs with thin tail bounds and makes SVT practical.

We provide the "Reuseable-Holdout" algorithm implemented with Gaussian Mechanism as follows.

Step 1  For each attribute $i \in [d]$ compute the correlation with the label on the training and holdout sets: $w_i^t = \sum_{(x,y) \in S_t} x_i y$ and $w_i^h = \sum_{(x,y) \in S_h} x_i y$.

Step 2  Sort $w_i^t$ with $k$ lagest values.

Step 3  For each of the $k$ features, if training correlation $|w_i^t| \geq 1/\sqrt{n}$; $|w_i^h| \geq 1/\sqrt{n}$, test if $|w_i^t - w_i^h| + \mathcal{N}(0, \sigma_2^2) > T$. If so, return $\perp$, else return $\top$.

Step 4  Pick out the subset of features (denoted as $V_k$) with output $\perp$ and construct a linear classifier $f(x) = \text{sign}(\sum_{i \in V_k} \text{sign}(w_i^t) \cdot x_i)$ using the sign of the training correlation.

In the Gaussian-SVT based algorithm, we first perturb $T$ with $\tilde{T} = T + \mathcal{N}(0, \sigma_1^2)$. Then we modefiy the "step 3" with testing if $|w_i^t - w_i^h| + \mathcal{N}(0, \sigma_2^2) > \tilde{T}$. If so, return $\perp$ and refresh $\tilde{T}$, else return $\top$.

We set $T = 0.04$ and $2\sigma_1 = \sigma_2 = 0.01$. As shown in Figure 4, we provide the average and standard deviation of results with 100 independent trials. Noting there are no correlations between $x$ and $y$, no classifiers can achieve accuracy better than 50%. However, a standard holdout (denoted as "training") results in accuracy $\gg 50\%$, which is overfitting. "Fresh" refers to the classifier accuracy on another fresh data of size $n$, which results in an accuracy 50%. Both the Gaussian mechanism and Gaussian SVT can prevent the algorithm from overfitting to the holdout set. Moreover, with the same noise scale to perturb queries, Gaussian-SVT pays privacy cost proportional to the size of features that outputs $\perp$. Hence it can answer exponentially more queries with the same privacy budget.

## H  One example for SVT with unbounded length

We provide an example for Laplace SVT with unbounded length as follows.

**Proposition 26.** Suppose we have an infinite sequence of queries $q_1, ..., q_\infty$ with ground truth at 0, threshold $T_i$ increases exponentially (i.e. $T_{i+1} = \theta T_i$) where $i$ is the index of the query. The Laplace noise $\nu_i \sim Lap(\lambda_2)$ is used to perturb each query. Then the expected length of SVT with a cut-off $c = 1$ is unbounded.

*Proof.* We first rewrite $T_0 = \beta \cdot \lambda_2$. Recall that if $Y \sim Lap(b)$, then $\Pr[Y \geq t \cdot b] = \frac{1}{2} \exp(-t)$. Therefore, we have $\Pr[K = 1] = \frac{1}{2} \exp(-\beta)$ where $K$ is a random variable indicating the stopping time. Moreover, $\Pr[K = 2]$ can be written as $\Pr\left[q_2 + \nu_2 \geq T_2 | q_1 + \nu_1 < T_1\right] \cdot \Pr[K \neq 1]$, which is smaller than $\frac{1}{2} \exp(-\beta \cdot \theta)$. Noting the probability of stopping at all integer can be written as

Figure 4: Learning uncorrelated label. The $x$-axis is the number of variables selected for the classifier. The $y$-axis indicates average classification accuracy over 100 executions.

$\lim_{i \to \infty} \Pr[K < i+1] = \sum_{i=1}^{\infty} \Pr[K = i] \leq \sum_{i=1}^{\infty} \frac{1}{2} exp(-\beta\theta^{i-1}) < 1$. Hence the expected length of SVT with a cut-off $c = 1$ is unbounded. $\qquad\square$