[Reviews · NeurIPS 2020]

Review 1

Summary and Contributions: This paper considers a generalization of the sparse vector technique for differential privacy, a classical technique used in privacy and adaptive data analysis. The proposed algorithm generalizes the classical one by allowing a user to chose the noise addition mechanism for it. The paper analyzes this algorithm using Renyi differential privacy and show several results for different choices of the parameters. The paper also compares experimentally the proposed algorithm (with different setup of the parameters) with the classical one.

Strengths: The proposed generalization allow one to parametrize the sparse vector technique in different dimension, and with different noise addition primitives, these seems to cover several cases that one can think about in different applications of this technique. The proposed generalization of the sparse vector technique is likely to be useful in practical applications of differential privacy, where small improvements can make the difference.

Weaknesses: From both the theoretical analysis and the experimental one, it seems that the difference in terms of performance between the Gaussian-SVT and the Laplace-SVT is quite limited. The privacy analysis seems essentially the same as the traditional one with few interesting observations which allow one to reason in RDP style.

Correctness: Several wrong variants of the sparse vector technique have appeared in the literature, and for this reason this technique has been the object of study for several works in the program verification community (under some assumption on the noise addition mechanism, the privacy of this technique can be proved in a automated way). As such, I think that the authors should have discussed more the correctness of their algorithm. Intuition suggests that the proposed generalization still enjoy the main properties that make the original algorithm private (mainly because it uses noise addition RDP mechanisms as parameters), and by looking at the proof in appendix it seems that it follow more or less the structure of the more traditional one, with some modification to support an analysis based on RDP. So, I am confident in the correctness of the results. Nevertheless, I think that the authors should discuss this in the paper.

Clarity: Overall, the paper is well structured and reads well. However, in some parts it seems that some general guideline would help a reader. For example, in section 3 there are several results and several remarks that try to explain interesting observations, but I find it rather confusing to go through them one after the other without some more general guideline.

Relation to Prior Work: The paper clearly discusses how this work differ from previous one. However, because of the history of the technique they consider, I think that they are missing some important related works. First, there are several generalizations of the SV technique, e.g. [1,2], and I think the paper should discuss whether it is likely that their generalization/analysis can be applied to those as well. Then, the paper missed the discussion about the correctness of the proposed analysis, as I already discussed in the correctness point. Some works that have looked at the correctness are [3,4,5]. [1] Mark Bun, Thomas Steinke, Jonathan Ullman: Make Up Your Mind: The Price of Online Queries in Differential Privacy. SODA 2017: 1306-1325 [2] Zeyu Ding, Yuxin Wang, Danfeng Zhang, Dan Kifer: Free Gap Information from the Differentially Private Sparse Vector and Noisy Max Mechanisms. Proc. VLDB Endow. 13(3): 293-306 (2019) [3] Gilles Barthe, Marco Gaboardi, Benjamin Grégoire, Justin Hsu, Pierre-Yves Strub: Proving Differential Privacy via Probabilistic Couplings. LICS 2016: 749-758 [4] Danfeng Zhang, Daniel Kifer: LightDP: towards automating differential privacy proofs. POPL 2017: 888-901 [5] Aws Albarghouthi, Justin Hsu: Synthesizing coupling proofs of differential privacy. Proc. ACM Program. Lang. 2(POPL): 58:1-58:30 (2018)

Reproducibility: Yes

Additional Feedback: I find confusing the way lemma 7 is formulated. You tie the resampling to the use of strong composition but it seems to me that strong composition could also be used for the algorithm where the resample option is set to false, am I missing something? The privacy proof in appendix B seems essentially the same as the proof for the classical version where you work directly with the privacy loss at alpha as required by RDP. The fictitious queries seems to be just a way to say that you focus on the worst case, and then you use a change of variable in the right steps. Is there any specific challenge you faced that I am missing? From the experiments in Fig 1 and Fig 2, it doesn't look like Gaussian SVT is always a reasonable choice in practice, or at least that the gain with respect to Laplace SVT is often negligible. Is there any specific setting where one should prefer always Gaussian SVT? ------------------------------------------ Comment after author feedback ------------------------------------------ I thank the author for addressing my comments in detail. I am more convinced now about the interest of the proposed mechanism and its proof. I have modified my score to reflect this.I suggest to the authors to integrate part of their response in the next version of the paper.


Review 2

Summary and Contributions: This paper considers a variation of the sparse vector algorithm, which is a classical algorithm in differential privacy. They show that they can improve on the original algorithm using Gaussian noise and with Renyi DP (a relaxation of traditional pure DP). They present a general framework of the sparse vector approach that would allow for plugging in various other private mechanisms.

Strengths: Within the differential privacy community, this is an interesting result in that it improves on a classical DP algorithm using some of the latest analytical tools from DP.

Weaknesses: The main limitation is that the result is pretty niche, in that it studies a particular algorithm in DP with different noise distributions at different steps while still providing DP guarantees. However, one of the limitations for actual deployments of DP to use sparse vector is due to easier mechanisms outperforming in most scenarios. This work might lead to more deployments using sparse vector. It is also an important algorithm in adaptive data analysis (which has a connection with DP).

Correctness: I skimmed the supplementary file and proofs and the general arguments make sense, i.e. no surprises.

Clarity: Can stand to have another read through to fix some typos.

Relation to Prior Work: Yes, it discusses the work in Renyi DP and sparse vector.

Reproducibility: Yes

Additional Feedback: This paper considers the classical Sparse Vector Technique from a practical lens. One aspect of making sparse vector more practical is to show that it can compete with existing privacy mechanisms with much fewer rounds. Gaussian noise has been found to have stronger privacy loss bounds when composed several times, using analysis based on Renyi divergence. So the natural question is whether there is a variant of sparse vector that uses Gaussian noise? An added benefit to using Gaussian noise is that the threshold parameter T need not be as large as with Laplace noise if the proportion of false positives is to be the same between the two noise additions. There seems to be a mistake in the parameter settings of eps_1 and eps_2 in Lemma 7 as the DP guarantee does not makes sense with those parameters. Introducing the stopping time random variable into the calculations of the RDP bound is interesting in that we can see the additional term in the privacy loss bound due to the Renyi analysis rather than the intuitive privacy loss bound that simply adds the privacy loss of both \rho and \nu DP mechanisms. Is this expectation on the stopping time necessary? From equation (3) it seems that Gaussian noise cannot be used for the mechanism that adds noise to the query values, but perhaps Gaussian noise can be used for both noise addition mechanisms and an infinite k_max can be used with a finite RDP bound. I am a little confused by the remark in lines 218-220. The work from Kairouz et al. ’15 gives the optimal DP composition of DP mechanisms. However, here you are obtaining an RDP bound and hence composition should be done with RDP. There need only be a single conversion of RDP to DP after composition. Is the remark saying that you are doing RDP -> DP in each round and then doing optimal DP composition on each one of these? Overall it is a nice result for a classical algorithm. Although it is presented as a general way to add noise, presumably only Laplace or Gaussian noise can be used. It also seems to answer an annoying issue with the classical SVT which required adding fresh noise to the threshold for each \top in order to get privacy loss that depends on \sqrt{c} rather than c. #### UPDATE ###### I have read the author feedback and my review remains the same. Thank you for addressing my comments!


Review 3

Summary and Contributions: In this work, the authors consider a stream of 1d threshold queries on a statistical dataset. The goal is to answer as many of these queries as possible under the constraint of differential privacy. The typical solution is called the sparse-vector technique (SVT) and uses the well-studied Laplace mechanism as a subroutine. The approach in this work is to consider a generalized form of SVT that uses other subroutines like the Gaussian mechanism. The paper contains experiments that show that the approach is competitive against existing work.

Strengths: The main contributions of this work are (1) the use of Renyi differential privacy (RDP) to analyze the general SVT algorithm and (2) experiments to justify swapping the Laplace mechanism with the Gaussian mechanism. As far as I am aware (1) is novel and the results of (2) are encouraging. The authors include the proper citations to related work. In broader terms, the results spur further work on SVT, RDP and adaptive data analysis.

Weaknesses: The experiments are fairly limited. Also, they do not explore the implications of their adaptive data analysis corollary. [the authors address this in the rebuttal]

Correctness: Yes

Clarity: Moderately so. There is a bit too much emphasis (in this submission version) on the very technical analysis while the experiments receive comparatively little attention.

Relation to Prior Work: Yes

Reproducibility: Yes

Additional Feedback:

[Author Response · NeurIPS 2020]

We thank all reviewers for the encouraging and helpful comments. All typos and minor comments will be fixed. Detailed responses are given below.

— — — — — — — — — — — — — — —**Reviewer 1**— — — — — — — — — — — — — — — — — ——

Thanks for your helpful suggestions and comments!

**» "General guideline fo section 3":** Thanks for the suggestion, we will revise our exposition of the results accordingly.

**» "More discussions on related works":** Thanks for sharing these related body of work that we missed. Re: [3,4,5], we will discuss these exciting progress in automating the DP proofs and the fact that SVT partially motivated them as in (Lyu et al., 2017). Re: [1,2]. Our technique should be directly applicable to the BetweenThreshold variant as in [1] and to also release the "gap" as in [2], but we need to take a closer look on [2] to ensure.

**» "challenges (and novelty) in the proof":** Thanks for checking the fine details of our proof. We acknowledge borrowing some ideas from the classical SVT analysis as we stated up front in the paper (Line 41). This particularly refers to the "change-of-variable" trick (which exists since [DNR+09,HR10]). However, the rest of the proof of Theorem 8 is quite different as we need to (1) bound the moments of the density ratio rather than it maximum; (2) formalize the *reduction* to the RDP of subroutines. The new and delicate arguments include:

1. The application of Jensen's inequality to bivariate joint-convex function $f(x,y) = x^\alpha y^{1-\alpha}$ (line 329).
2. The "fictitious query" argument (which is used in stating and proving Lemma 17) is actually important in formalizing the reduction.
3. The stopping time random variable and the use the alternative definition of expectation (line 366-374) are new.
4. Lemma 18 is somewhat cute too, in how it handles the case with $\infty$.

**» "Is there any specific setting where one shall always prefer Gaussian SVT?"**

Thanks for the thoughtful question. Gaussian SVT is not always preferred over Laplace SVT due to many different dimensions in comparison. But in many cases, it could work better in practice. Our observation is that, when the tail of the noise plays a significant role, e.g. the threshold $T$ is large (Figure $b$ in Experiment 1), Gaussian-SVT is more advantageous due to a more concentrated noise. To further improve Gaussian-SVT, the stage-wise Gaussian-SVT that uses hybrid composition outperforms Laplace-SVT significantly.

**» Question on lemma 7:** Lemma 7 is a combination of Theorem 3.25 from (Dwork and Roth, 2014) and Theorem 2 from (Lyu et al., 2017). For "strong composition"-style result, we were referring to a SVT with cut-off 1 where the dependence on $c$ in the final bound is $O(\sqrt{c})$. To the best of our knowledge, "non-resampling" versions of SVT has not been shown to achieve such a bound, except in restricted settings (from our Theorem 11).

— — — — — — — — — — — — — — —**Reviewer 4**— — — — — — — — — — — — — — — — — ——

Thank you for your helpful comments!

**» "There seems to be a mistake in the parameter settings of $\epsilon_1$ and $\epsilon_2$ in Lemma 7."**

Thanks for spotting the typo, it shall be $\frac{\triangle}{\epsilon_1} = \frac{\sqrt{32c\log(1/\delta)}}{\epsilon}$.

**» "Is the expectation on the stopping time necessary?"**

This is an excellent observation. We have good reasons to believe that it is necessary unless pure-DP subroutines are used. A lower bound would be nice, but is beyond the scope of this paper. We focused on further bounding that conditional moments involving the stopping time RV by considering a non-negative query class (see Line 156-170).

**» Remark in lines 218-220: doing RDP to DP in each round and then doing optimal DP composition on each one of these?** The reviewer might have missed the motivation of the stage-wise approach. Direct composition of the RDP bound in Theorem 8 with a single conversion to $(\epsilon, \delta)$-DP does not allow a $O(\sqrt{c})$ scaling unless $\delta$ is very small, see our discussion in line 188-199. Line 218-220 is a comment about the choice of $c', k'$ in Theorem 12 and 13.

**» "Only Laplace or Gaussian noise can be used?"** Any noise-adding procedures that admit an RDP bound can be used, e.g., the optimal stair-case noise from (Geng and Viswanath, 2014).

— — — — — — — — — — — — — — —**Reviewer 5**— — — — — — — — — — — — — — — — — ——

Thanks for your kind comments! We will add experiments related to adaptive data analysis following [DFHPRR-14] in the appendix to demonstrate the merits of Gaussian-SVT. Notice that they added Gausssian noise, rather than the Laplace noise they analyzed, so we are confident that the experimental results will be to our favor.

[Meta-Review · NeurIPS 2020]

This paper proposes a generalization of a basic technique in differential privacy, known as the sparse vector technique. The paper offers a new style of privacy analysis for the proposed technique using Renyi Differential Privacy. The proposed technique allows for plugging in other private mechanisms as opposed to the Laplace mechanism used in the original version of the sparse vector technique. The paper also contains several experiments showing the competitiveness of the proposed technique against existing algorithms. This is an interesting result that improves and extends a basic technique in differential privacy, and could spur further work.